# Identification of NAD-RNA species and ADPR-RNA decapping in Archaea

José Vicente Gomes-Filho [1]✉, Ruth Breuer[1], Hector Gabriel Morales-Filloy[2], Nadiia Pozhydaieva[3], Andreas Borst[4], Nicole Paczia [3], Jörg Soppa [4], Katharina Höfer[3,5], Andres Jäschke [2] & Lennart Randau [1,5]✉

NAD is a coenzyme central to metabolism that also serves as a 5′-terminal cap for bacterial and eukaryotic transcripts. Thermal degradation of NAD can generate nicotinamide and ADP-ribose (ADPR). Here, we use LC-MS/MS and NAD captureSeq to detect and identify NAD-RNAs in the thermophilic model archaeon *Sulfolobus acidocaldarius* and in the halophilic mesophile *Haloferax volcanii*. None of the four Nudix proteins of *S. acidocaldarius* catalyze NAD-RNA decapping in vitro, but one of the proteins (Saci_NudT5) promotes ADPR-RNA decapping. NAD-RNAs are converted into ADPR-RNAs, which we detect in *S. acidocaldarius* total RNA. Deletion of the gene encoding the 5′−3′ exonuclease Saci-aCPSF2 leads to a 4.5-fold increase in NAD-RNA levels. We propose that the incorporation of NAD into RNA acts as a degradation marker for Saci-aCPSF2. In contrast, ADPR-RNA is processed by Saci_NudT5 into 5′-p-RNAs, providing another layer of regulation for RNA turnover in archaeal cells.

The discovery of NAD, a cofactor critical to cellular metabolism, as a 5′ cap in bacteria challenged earlier notions that only eukaryotes utilize RNA capping mechanisms[1]. Since the first discovery of NAD−RNA caps, additional reports of their presence in Gram-positive bacteria and eukaryotes such as *Arabidopsis thaliana*, *Saccharomyces cerevisiae*, and mammalian cells suggest that this RNA modification is ubiquitous in the Tree of life[2–8]. Mechanistic studies demonstrated that bacterial RNA polymerase (RNAP) and eukaryotic RNAP II can utilize NAD, NADH, flavin adenine dinucleotide (FAD), adenosine diphosphate ribose (ADPR), and 3′-dephospho-coenzyme A (dpCoA) to initiate transcription at promoters containing A at its +1 position[4,9,10]. Additionally, NAD-caps on mammalian small nucleolar RNAs (snoRNAs) and the related small Cajal body RNAs (scaRNAs) suggest an additional post-transcriptional capping mechanism in eukaryotic cells[6].

In human and fungal cells, the non-canonical decapping enzymes DXO/Rai1 are responsible for initiating NAD−RNA degradation by removing the NAD-cap[3,6]. In *Escherichia coli*, NAD decapping is performed by a nucleoside diphosphate linked to another moiety X (NUDIX) hydrolase termed NudC[11]. This enzyme hydrolyzes the NAD-cap resulting in nicotinamide mononucleotide (NMN) and a 5′-monophosphate RNA (5′-p-RNA) that is efficiently degraded by cellular endonucleases[8,12,13]. Further studies aiming to elucidate the function of NAD−RNAs revealed striking differences in the roles of this modification in bacterial and eukaryotic cells. In *E. coli*, this modification was initially thought to protect the RNA against pyrophosphohydrolase (RppH) and RNase E degradation. However, recent in vitro studies identified RppH as an additional NAD-decapping enzyme[12]. In *Bacillus subtillis*, it was shown that NAD modification of RNAs prevents 5′ → 3′ exonucleolytic activity from RNase J1, suggesting a stabilizing role[8]. On the other hand, in eukaryotic cells, NAD-caps are considered to promote RNA decay[6], and a highly efficient surveillance machinery for the degradation of NAD−RNAs was described for yeast[13]. NAD-caps can be related to different biological outcomes, even in organisms from the same domain of life, as demonstrated by the putative translational capacity of NAD−RNAs in eukaryotic cells[5,6]. Moreover, the 5′ → 3′ exonucleases Xrn1 and Rai1 from yeast are suggested to directly influence the concentration of free NAD by releasing intact NAD from NAD−RNAs[13].

[1]Faculty of Biology, Philipps-Universität Marburg, Marburg, Germany. [2]Institute of Pharmacy and Molecular Biotechnology (IPMB), Heidelberg University, Heidelberg, Germany. [3]Max Planck Institute for Terrestrial Microbiology, Marburg, Germany. [4]Institute for Molecular Biosciences, Biocentre, Goethe-University, Frankfurt am Main, Germany. [5]SYNMIKRO, Center for Synthetic Microbiology, Marburg, Germany. ✉e-mail: gomesfil@staff.uni-marburg.de; randau@staff.uni-marburg.de

The degradation of NAD at high temperatures (>75 °C) into nicotinamide (Nm) and ADPR demands hyperthermophilic microorganisms to present robust pathways for salvaging and preventing the putative glycation and glycoxidation of proteins[14–16]. In mesophilic organisms, the generation of ADPR is mainly achieved through enzymatic reactions performed by enzymes such as ADPR-transferases, cyclic ADPR-synthases, and poly ADPR polymerases[16]. A recent study provided the first evidence for 5′ ADP-ribosylated RNAs in mammalian cells[17]. Interestingly, the process of ADPR-capping in eukaryotes has different pathways. The human protein CD38, for example, can convert NAD–RNA to ADPR–RNA by removing Nm from the NAD[18]. The bacterial RNA 2′-phosphotransferase (Tpt1) and its orthologues from higher organisms, TRPT1, can use free NAD to ADP-ribosylate 5′-p-RNA substrates[17]. In both cases, contrary to NAD-capping, the generation of ADPR–RNAs is achieved post-transcriptionally. Furthermore, ADPR–RNAs were shown to be more resistant to Xrn1 exonuclease activity while not supporting translation[17].

As we gain insights into the functional consequences of NAD-capping in bacteria and eukaryotes, this information is lacking for archaea. Here, we combine LC–MS/MS and NAD captureSeq methodologies to detect and identify NAD-capped RNAs in the archaeal model organisms *Sulfolobus acidocaldarius* and *Haloferax volcanii*. Multiple NUDIX family proteins can be involved in processing mRNA caps[19]. A sequence similarity search provided four NUDIX hydrolases in *S. acidocaldarius*. In vitro assays using recombinant enzymes did not

reveal NAD decapping activity. Instead, we detected that SACI_RS00060 (here renamed to Saci_NudT5) showed activity following heat exposure of NAD-capped RNAs in vitro. We show that the thermal degradation of NAD–RNAs generates ADPR–RNA in vitro and that these transcripts are present in *S. acidocaldarius'* total RNA. Furthermore, we provide evidence that the knockout of a 5′–3′ exonuclease (Saci-aCPSF2) impacts NAD–RNA concentration and suggests that NAD-capping influences RNA turnover in *S. acidocaldarius*.

## Results

### Detection of NAD-capped RNAs in *S. acidocaldarius* and *H. volcanii*

First, we aimed to determine NAD modifications of RNAs in the crenarchaeon *S. acidocaldarius* and the euryarchaeon *H. volcanii*. Nuclease P1 is a 3′ → 5′ exonuclease that releases single nucleotides without affecting pyrophosphate bonds, leaving capping nucleotides, like NAD, intact after release (Fig. 1a). Total RNA was isolated from *S. acidocaldarius* and *H. volcanii* and treated with nuclease P1. Identification and relative quantification of NAD in the nuclease P1 treated isolates were done using LC–MS/MS (Fig. 1b, c, black lines). To determine the background levels of co-purified free NAD, we analyzed RNA treated with heat-inactivated nuclease P1 (Fig. 1b, c, red lines). The specific increase of relative NAD levels upon treatment with active nuclease P1 indicated the presence of NAD–RNAs in both *S. acidocaldarius* and *H. volcanii*.

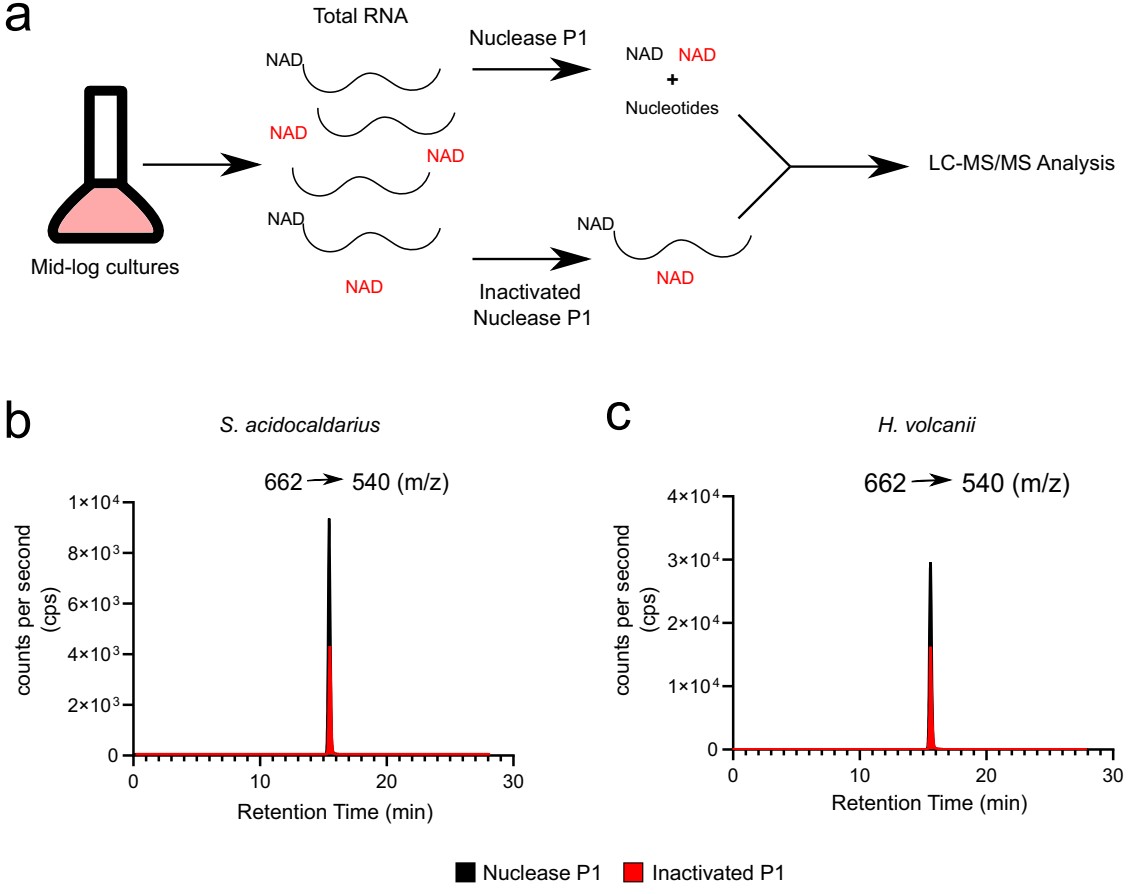

**Fig. 1 | Detection of NAD in total RNA extract from *S. acidocaldarius* and *H. volcanii*. a** Method utilized to detect free (red) or RNA-bound NAD (black). Briefly, total RNA is extracted from mid-log cultures and digested with either nuclease P1 or a heat-inactivated enzyme. Next, samples are submitted to LC–MS/MS analysis, and NAD is measured. **b** Extracted ion chromatogram of the NAD specific mass transition 662 (m/z) → 540 (m/z) for *H. volcanii* total RNA digested with nuclease P1

(Black), inactivated nuclease P1 (Red). **c** Extracted ion chromatogram of the NAD specific mass transition 662 (m/z) → 540 (m/z) for *S. acidocaldarius* total RNA digested with nuclease P1 (Black), Inactivated nuclease P1(Red). **b**, **c** represent results from one of three independently performed experiments with similar results. Source data are provided as a Source Data file.

## Identification and classification of NAD−RNAs in *S. acidocaldarius* and *H. volcanii*

To obtain a snapshot of the NAD−RNA populations from both organisms, NAD captureSeq was performed in the mid-log growth phase (Supplementary Fig. 1a, b)[1, 12]. The libraries were sequenced on an Illumina HiSeq3000, and at least 6 million reads were obtained per sample. The obtained reads were trimmed and aligned against the archaeal genomes of interest. Next, DESeq2[20] was used to determine enriched transcripts (p-adjusted value < 0.1 and log2(Fold Change) ≥ 1) in samples treated with an ADP-ribosyl cyclase from *Aplysia california* (ADPRC+) versus non-treated samples (ADPRC-)[1,12]. Using these threshold values, we identified 86 NAD−RNAs for *H. volcanii* and 83 NAD−RNAs for *S. acidocaldarius* (Supplementary Data 1 and 2, Supplementary Fig. 1c–f). To validate the *S. acidocaldarius* NAD capture-Seq results, qPCR analysis was performed with cDNA obtained after ligation of the second adapter. This experiment had two NAD captureSeq enriched genes, *tfb*, and SACI_RS10480, and one negative control, SACI_RS03345, as targets. In agreement with the enrichment detected by NAD captureSeq, *tfb*, and SACI_RS10480 showed a relative expression of 25 ± 10 and 45 ± 20, respectively. The negative control SACI_RS03345 showed no enrichment (Supplementary Fig. 1g). APB northern blot analysis[21] of selected targets (Sac_62_asRNA and Sac_6_k-turn) was not sensitive enough to obtain clear bands for the modified transcripts (data not shown), pointing to the low abundance of individual RNAs. Thus, we applied an enrichment-based method for the detection of NAD-caps in selected transcripts. Briefly, we submitted total RNA from *S. acidocaldarius* (spiked-in with a NAD-model-RNA positive control) to qRT-PCR after ADPRC-catalyzed biotinylation of NAD−RNAs followed by streptavidin bead enrichment[22]. Significant enrichment was verified for seven transcripts in the biotinylated fraction (ADPRC+) (Supplementary Fig. 1h). Next, we compared the 50 most abundant transcripts in the ADPRC+ libraries with the 50 most abundant transcripts in a published small RNA sequencing (sRNA-seq) library obtained under identical growth conditions[23]. Of the enriched RNAs, only six were among the most expressed in *S. acidocaldarius*, reinforcing a selective enrichment of NAD−RNAs rather than a bias for overly abundant transcripts (Supplementary Data 3 and Supplementary Fig. 2). Analysis of the nucleotide frequency of the +1 NAD transcription start sites (NAD−TSS), and the −1 positions demonstrated that all enriched transcripts start with an adenine (Fig. 2a, b). For both *H. volcanii* and *S. acidocaldarius*, the −1 position was enriched for thymine. For positions −2 to −3, *S. acidocaldarius* presented slight enrichment for A/T compared to G/A in *H. volcanii* (Fig. 2a, b). To further evaluate if the addition of the NAD-cap occurs co-transcriptionally, we analyzed the upstream regions (−50 bp) of the identified NAD−RNAs for recognizable promoter motifs. A TFB recognition element (BRE) was detected at around position −30 for *S. acidocaldarius*. A TATA box motif was also detected at around position −26 for *S. acidocaldarius* and position −28 for *H. volcanii* (Fig. 2a, b)[24,25]. Next, we sought to compare the NAD−TSS with the primary transcription start sites (pTSS) for RNAs with a 5′-ppp termini. To this end, we prepared dRNA-Seq libraries for *S. acidocaldarius* (Supplementary Data 4, Supplementary Fig. 3) and reanalyzed published data for *H. volcanii*[26]. Manual curation of the positions showed that most NAD−TSS and pTSS are found at the same positions (76% for *S. acidocaldarius* and 90% for *H. volcanii*) (Fig. 2c, Supplementary Data 1 and 2). The high number of overlapping NAD-TSSs and pTSSs, together with the detection of distinct promoter motifs, points to the possibility that the enriched RNAs might be co-transcriptionally capped with NAD[9,27].

To explore potential patterns of functional enrichment of capped RNA molecules, the identified NAD−RNAs were divided into five categories: (I) Internal: the +1 position is located within a coding gene; (II) Start codon: the +1 position matches the annotated start codon; (III) tRNAs; (IV) small RNAs (sRNAs, e.g., C/D box sRNAs and non-coding sRNAs); (V) 5′ UTRs: the +1 position is located upstream of the start codon of an enriched coding gene (Fig. 2d, e). Comparing the abundances of each class between *H. volcanii* and *S. acidocaldarius* revealed some striking differences. First, nine tRNAs were enriched in *S. acidocaldarius*, while only tRNA-Met was enriched in *H. volcanii* (Supplementary Data 1 and 2). Second, only 7% of the enriched mRNAs had a NAD−TSS that matched the start codon adenosine of the respective genes, as opposed to 41% in *H. volcanii*. Moreover, the analysis of our dRNA-seq data (Supplementary Fig. 3a–c) demonstrates an abundance of pTSS at the first base of the start codon (Supplementary Fig. 3b), supporting earlier studies that indicated leaderless transcription to be a common feature in Haloarchaea and Sulfolobales[26, 28].

The number of NAD-capped sRNAs detected in *S. acidocaldarius* was almost four times higher than for *H. volcanii*. Next, to obtain an overview of the enriched gene functions, archaeal clusters of orthologous genes (arCOGs) were used to group genes according to different biological functions[29]. However, no clear enrichment of specific arCOGs was detected (data not shown).

## NUDIX proteins from *S. acidocaldarius* have ADPR-decapping activity but cannot perform NAD-decapping

In bacteria, the first identified NAD decapping enzyme was NudC, a member of the NUDIX family, which hydrolyzes the NAD-cap resulting in 5′-p-RNA and free NMN[11]. The family of Nudix hydrolases encompasses functionally diverse and versatile proteins, all containing the conserved Nudix motif with the consensus sequence $GX_5EX_5U/AXREX_2EEXGU$ (U for hydrophobic residue and X for any residue)[19]. More recently, both *E. coli* RppH and *Bacillus subtilis* BsRppH were also shown to perform in vitro NAD decapping in addition to their pyrophosphohydrolase activities[8]. A bioinformatic search for potential Nudix hydrolases in *S. acidocaldarius* yielded four protein candidates (Fig. 3a, Supplementary Data 5). All four candidate proteins (SACI_RS00060, SACI_RS00575, SACI_RS00730, and SACI_RS02625) possess the conserved glutamic acid residues in the Nudix motif, which are crucial to the hydrolase activity[11,19,30–32]. Another notable feature is the residue at position 16 following the G of the Nudix motif. The residue at this position was shown to suggest a possible substrate for the respective Nudix protein and therefore serves to identify and distinguish different subsets of Nudix hydrolases[32]. In SACI_RS00060, a proline at this position and grouping with other known ADPRases (Fig. 3b) suggests ADPR hydrolysis activity, while in SACI_RS00575, the tyrosine hints at activity towards polyphosphate dinucleoside substrates[32]. For SACI_RS00730 and SACI_RS02625, no residue pointing at a specific activity was identified.

Next, combining heterologous expression in *E. coli* and in vitro cell-free protein synthesis, we produced and purified the identified Nudix proteins and generated individual Nudix domain mutants (NDM) (Supplementary Fig. 4). To evaluate the NAD decapping activity of these proteins, a synthetic RNA (Model-RNA), containing a single A at its transcription start site, was in vitro transcribed using NAD, GTP, CTP, and UTP. Replacing ATP with NAD ensures that the in vitro transcription reaction only initiates with the latter, providing pure NAD−RNA substrates. None of the recombinant *S. acidocaldarius* Nudix proteins performed NAD decapping in vitro (Fig. 3c), suggesting that *S. acidocaldarius* has no enzymatic NAD decapping activity or an enzyme other than a Nudix hydrolase is responsible for this process. As NAD was shown to be converted into ADPR and Nm at higher temperatures[14], we evaluated if any *S. acidocaldarius* recombinant Nudix protein could hydrolyze ADPR-RNA instead of NAD−RNAs. It was previously demonstrated that the Human NudT5 (hNudT5) hydrolyzes free ADPR in vitro[31]. Additionally, through sequence analysis, SACI_RS00060 (here renamed to Saci_NudT5) clusters with other known ADPR-hydrolases, including hNudT5 (Fig. 3b). This prompted us to test the activity of these proteins against ADPR−RNAs. To this

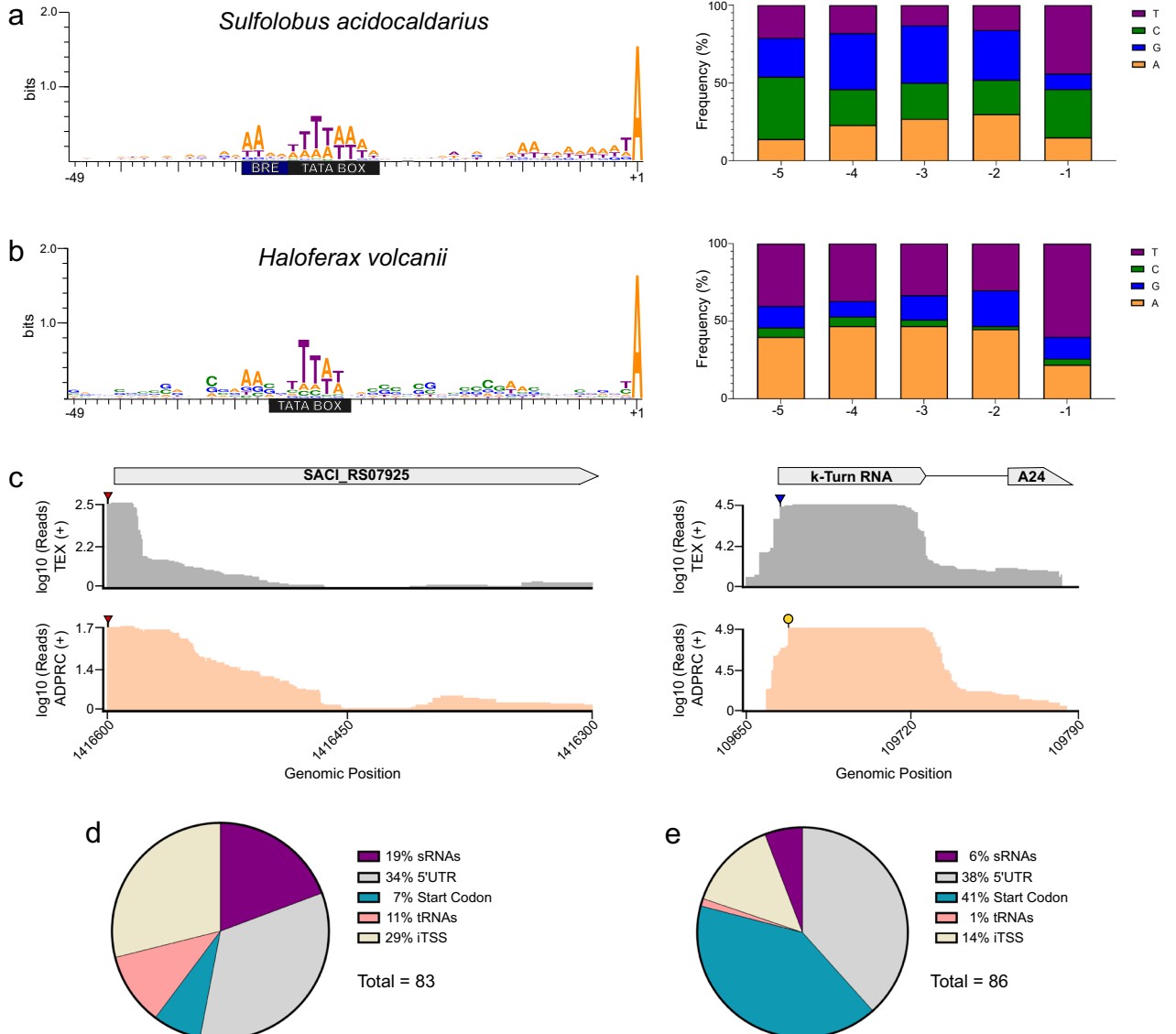

**Fig. 2 | Promoter identification, nucleotide frequency at the −1 to −5 positions for NAD−RNAs, and comparison with primary transcription start sites (pTSS). a** Promoter and nucleotide frequency analysis for *S. acidocaldarius*. The blue rectangle represents the TFB recognition element (BRE) and the black rectangle represents the TATA box motif. **b** Promoter and nucleotide frequency analysis for *H. volcanii*. The black rectangle represents the TATA box motif. **c** Comparison of transcription start sites identified by dRNA-Seq (gray lines) and NAD captureSeq (salmon lines). Left panel: Coverage plot of carboxypeptidase M32 (SACI_RS07925) with matching NAD− and pTSS (red triangles). Right panel: Coverage plot of a k-turn RNA upstream of the peptidase A24 with non-matching NAD−TSS (yellow circle) and pTSS (blue triangle). Classification of NAD−RNAs identified in *S. acidocaldarius* (**d**) and in *H. volcanii* (**e**). Source data are provided as Source Data file.

end, pure ADPR−RNA substrates were generated as described above for NAD−RNAs by exchanging NAD for ADPR in the in vitro transcription reaction. The application of ADPR-decapping assays revealed that Saci_NudT5 could convert ADPR−RNAs to 5′-p-RNAs (Fig. 3d). Total RNA from mid-log cultures of both *S. acidocaldarius* MW001 (WT) and ΔSaci_NudT5[33] was treated with nuclease P1 and submitted to LC−MS/MS for the detection of NAD and ADPR. For the WT strain, $2 \pm 0.3$ fmol of NAD and $16 \pm 1$ fmol of ADPR were detected per µg of RNA. On the other hand, the ΔSaci_NudT5 strain showed $0.3 \pm 0.06$ fmol of NAD and $4.43 \pm 0.05$ fmol of ADPR per µg of RNA indicating a fold-change of $0.22 \pm 0.08$ for NAD and $0.35 \pm 0.03$ for ADPR when comparing ΔSaci_NudT5 to the WT (Supplementary Fig. 5a, b, Supplementary Table 1). This observed increase in the ratio of ADPR/NAD suggests that both the incorporation of NAD to RNA and the ADPR-decapping activity of this strain might be reduced. Preparation of S30 cell extracts from both strains and incubation with ADPR−RNAs evidenced that the knockout strain presents a slight decrease in ADPR-decapping activity

(Supplementary Fig. 5c, d). These combined results indicate that the knockout of Saci_NudT5 significantly reduces the amount of NAD and ADPR−RNAs and suggests the presence of additional enzymes involved in ADPR-decapping.

As *S. acidocaldarius* might lack proteins similar to the human CD38 and the TIR domain proteins from bacteria[18,34] (Supplementary Data 6), protein families (Pfam) that are associated with NAD-consuming enzymes were used for an HMMER analysis[35]. We detected nine potential NADases that can be future targets for studying NAD metabolism in *S. acidocaldarius* (Supplementary Table 2).

## NAD−RNAs are converted to ADPR−RNAs by thermal degradation

In thermophilic environments (>60 °C), such as the natural habitats of *S. acidocaldarius*, NAD is quickly degraded into ADPR and Nm, the first having the potential to be toxic by spontaneously reacting with proteins[14,16,36]. Besides, hyperthermophilic archaea contain robust

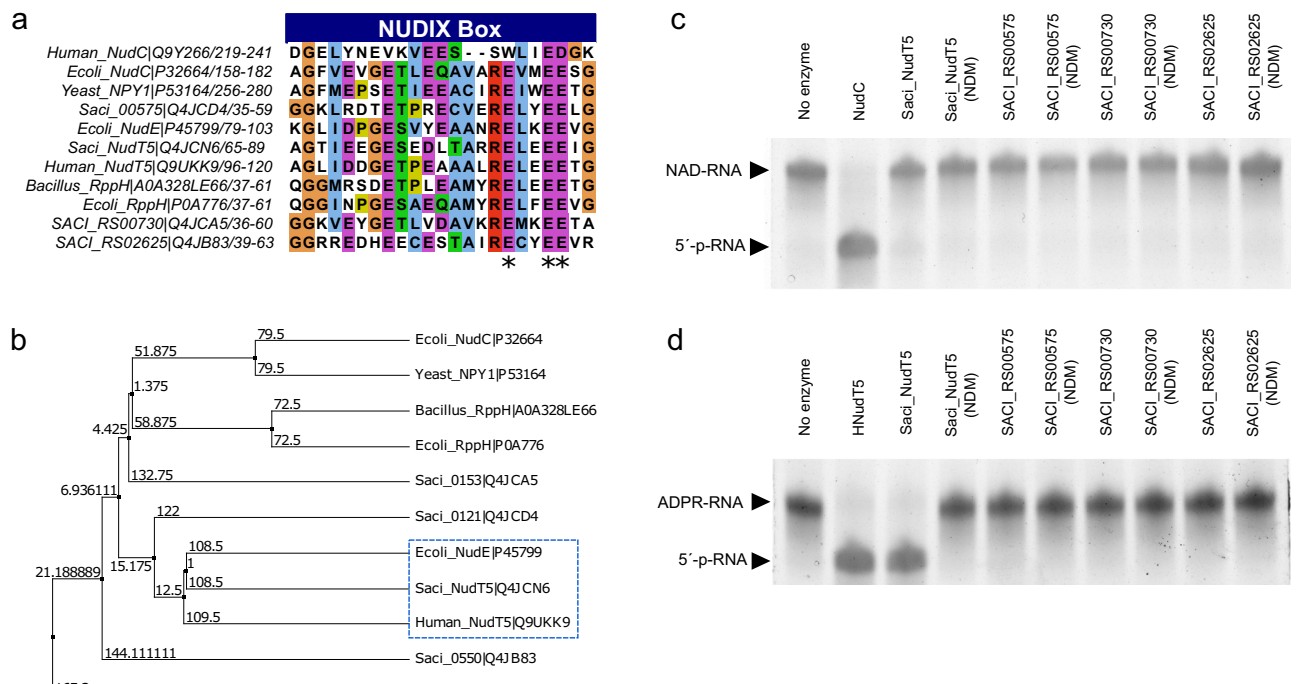

**Fig. 3 | Identifying Nudix proteins in *S. acidocaldarius* and evaluating NAD and ADPR-decapping activity. a** Alignment of Nudix proteins of *S. acidocaldarius* and other organisms (Supplementary Data 5). Blue rectangle: Nudix motif. Asterisks: amino acids selected to obtain Nudix domain mutants (NDM) for each protein. Amino acids are colored according to the ClustalX color scheme (Light blue: Hydrophobic and >60% site occupancy; Red: Positively charged and >60% site occupancy; Magenta: Negatively charged and >60% site occupancy; Green: Polar and >60% site occupancy; Orange: Glycines; Yellow: Prolines. **b** Average distance tree using BLOSUM62 showing the grouping of Saci_NudT5 with the previously described ADPR-hydrolases NudE and hNudT5[18,55]. **c** NAD decapping activity of the four Nudix candidates and their respective NDM was evaluated in vitro and resolved on APB-gels. **d** ADPR-decapping activity of the four Nudix candidates and their respective NDM was evaluated in vitro and resolved on APB-gels. Saci_NudT5 performed ADPR-decapping, and Saci_NudT5 (NDM) lost this activity. **c, d** The results from one of three independently performed experiments with similar results. Source data are provided as a Source Data file.

pathways to salvage NAD from its degradation products and avoid their accumulation[14,15]. Thus, we performed thermal degradation experiments to interrogate the stability of NAD covalently linked to RNAs under pH and temperatures that mimic the *S. acidocaldarius* habitat and physiology (cytoplasmic pH: 6.5 and growth temperature 75 °C)[14,37]. Briefly, in vitro transcribed model NAD–RNA was incubated at 75 °C or 85 °C in the presence of 50 mM Tris-HCl buffer (pH 6.5 at 75 °C or pH 6.5 at 85 °C) for up to 2 h. To track the conversion of NAD–RNA into ADPR–RNA, the heat-treated NAD–RNA was used for an ADPR-decapping assay with hNudT5, which shows in vitro activity toward ADPR but not NAD–RNAs[18]. This assay evidenced specific conversion of NAD–RNA to ADPR–RNA after exposure to high temperatures (Fig. 4a, b). To further analyze this reaction, NAD–RNAs were heated to 85 °C in 50 mM Tris-HCl buffer (pH 6.5 at 85 °C) for up to 2 h and digested with nuclease P1. To establish the half-life of NAD covalently linked to RNA, NAD and its respective degradation products, Nm and ADPR, were quantified by LC-MS/MS (Fig. 4c). The obtained half-life of 31 min suggests that NAD covalently linked to RNA remains heat-labile (Fig. 4d) generating heat degradation products that can be recovered by NAD salvage pathways[14].

**Deletion of the *S. acidocaldarius* 5′-3′ exonuclease Saci-aCPSF2 alters the ratio of NAD–RNAs and ADPR–RNAs**

RNase J is a widespread exo/endoribonuclease in bacteria and archaea[38]. In *B. subtillis*, the 5′-3′-exonucleolytic activity of RNase J1 relies on a monophosphate group at the 5′ end of different transcripts[8]. The NAD-cap was reported to be less efficient in stabilizing RNA against RNase J1 activity than a 5′-ppp terminus[8]. In *S. acidocaldarius*, the RNase J orthologue Saci-aCPSF2 was shown to act as an exonuclease against 5′-p-RNAs substrates while retaining some activity against 5′-ppp-RNAs[39,40]. To evaluate the impact of this exonuclease on

NAD–RNA levels, a knockout strain of Saci-aCPSF2 was generated. In agreement with previous reports[40], no growth defect was detected for this strain (data not shown). Next, total RNA from mid-log cultures of both *S. acidocaldarius* MW001 (WT) and ΔSaci-aCPSF2 (KO) was incubated with nuclease P1 and submitted to LC-MS/MS for the detection of NAD and ADPR. In the KO strain, 8.7 ± 0.6 fmol of NAD and 20 ± 1.2 fmol of ADPR were detected, these values result in an 8:1 ratio of ADPR/NAD released from total RNA in the WT and a 2:1 ratio for the KO (Fig. 5a). A comparison between strains evidenced 4.5-fold more NAD released from total RNA in the KO, pointing to an accumulation of capped transcripts in this genetic background (Fig. 5b). ADPR, on the other hand, presented only a slight 1.25-fold increase (Fig. 5b, Supplementary Table 1). Next, to interrogate the nuclease and decapping activities of the WT and KO strains, S30 cell extracts were prepared and incubated with in vitro transcribed NAD- and ADPR-model-RNA substrates. Surprisingly, when comparing the activity of the WT S30 cell extracts with the KO S30 extract, we observed that NAD–RNAs present a faster degradation rate in the WT strain (Fig. 5c, d). In both cases, ADPR–RNA was processed into 5′-p-RNA (Fig. 5e, f). Altogether, these results suggest that the 5′-3′exonuclease Saci-aCPSF2 plays an essential role in the turnover of NAD-capped RNAs.

## Discussion

NAD and related dinucleotide metabolites are essential for many physiological processes, and their detection as 5′ caps for different bacterial and eukaryotic RNAs revealed additional layers of complexity[41]. Nevertheless, the specific roles of NAD-caps are still being uncovered.

In the present study, the detection and characterization of NAD–RNAs in the crenarchaeon *S. acidocaldarius* and the euryarchaeon *H. volcanii* provided evidence that NAD-capping of RNA

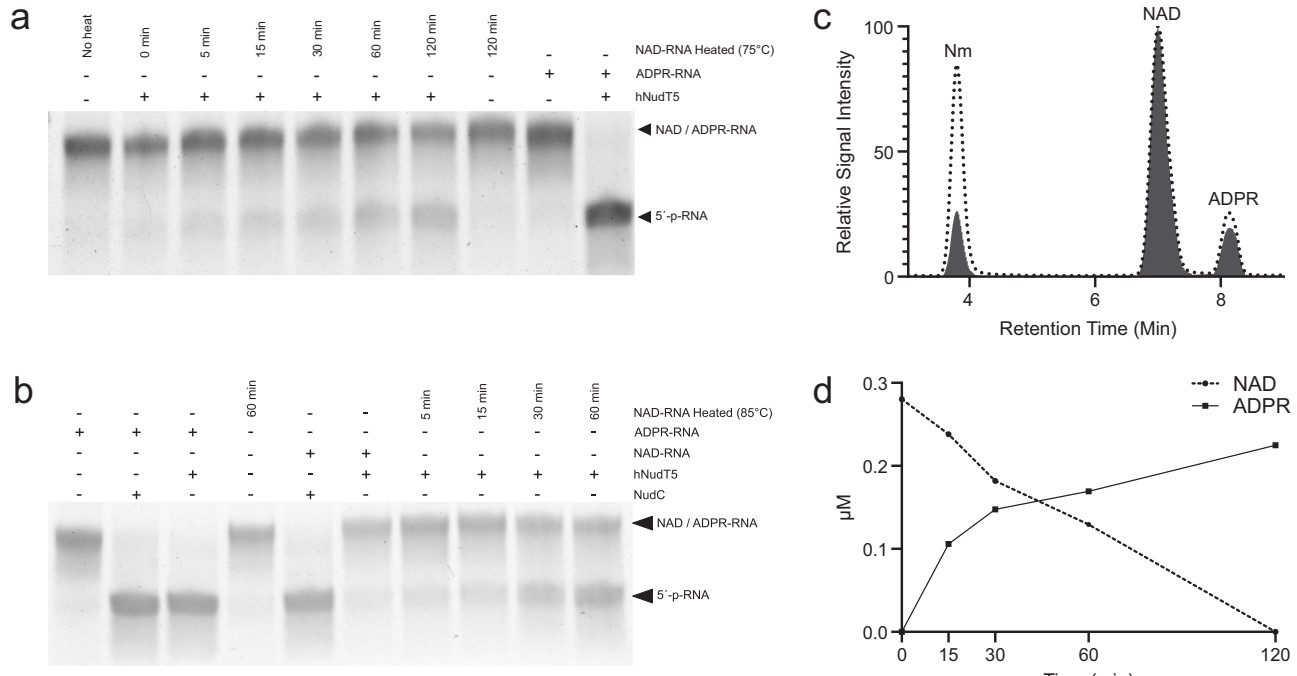

**Fig. 4 | NAD–RNAs are converted to ADPR–RNAs by thermal degradation.**
**a** NAD–RNAs were incubated at 75 °C for up to 120 min in 50 mM Tris-HCl (pH 6.5 at 75 °C). The reaction products were then incubated with hNudT5, and the conversion to 5′-p-RNA was monitored with APB-gels. In vitro transcribed NAD–RNA and ADPR–RNA were used as controls for hNudT5 reactions. **b** NAD–RNAs were incubated at 85 °C for up to 60 min in 50 mM Tris-HCl (pH 6.5 at 85 °C). The reaction products were then incubated with hNudT5, and the conversion to 5′-p-RNA was monitored with APB gels. In vitro transcribed NAD–RNA and ADPR–RNA were used as controls for hNudT5 and NudC reactions. **c** Extracted ion chromatogram of chemically pure Nm, NAD, and ADPR standards (gray area) and a heat-treated sample of NAD–RNA (30 min) (dotted lines). **d)** NAD–RNAs were incubated at 85 °C for 5, 15, 30, 60, and 120 min in 50 mM Tris-HCl (pH 6.5 at 85 °C), digested with nuclease P1 and submitted to LC–MS/MS analysis for the quantification of NAD and ADPR. The $t_{1/2}$ of NAD covalently linked to RNA was obtained with a typical decay equation ($dC/dt = -kC$). The calculated half-life was 31 min. **a, b** The results from one of three independently performed experiments with similar results. Source data are provided as a Source Data file.

molecules is common to all domains of life. In both archaea, an enrichment of the Transcription Initiation Factor IIB (TFIIB) was observed, arguing for a conserved role of the NAD-cap in this gene's transcript. Moreover, in *H. volcanii*, the mRNA of a NAD-dependent protein deacetylase gene from the SIR2 family has been identified to be NAD-capped, suggesting a possible connection between the intracellular levels of free NAD and NAD capping. In eukaryotic cells, the biogenesis of NAD-capped tRNAs and snoRNAs is still a point of contention. A previous study demonstrated that some NAD-capped snoRNAs and tRNAs did not possess a recognizable upstream sequence motif that supports NAD-initiated transcription[42]. Another recent report indicated that NAD caps could be added post-transcriptionally in eukaryotic cells[43]. In our *S. acidocaldarius* dataset, the detected pre-tRNAs and eight of eleven C/D box sRNAs present the same NAD-TSS and pTSS (Supplementary Data 1), suggesting that NAD capping most likely occurs co-transcriptionally for these transcripts. Additionally, we offer the first report of ADPR–RNAs in prokaryotes as described for *S. acidocaldarius*. None of the recombinant Nudix proteins from *S. acidocaldarius* exhibited NAD decapping activity, suggesting that this organism might utilize different pathways to process NAD–RNAs. Previous reports demonstrated that the non-canonical decapping enzymes DXO/Rai1 release intact NAD molecules from NAD–RNAs[3,6,13]. The highly conserved 5′-monophosphate 5′-3′ exoribonucleases, Xrn1 and Rat1, and their interacting partner Rai1 can also associate and hydrolyze NAD–RNAs in vitro[13]. A previous study in *B. subtillis* showed that the presence of a NAD-cap can reduce the 5′-3′ exonucleolytic activity from RNase J1[8]. In *S. acidocaldarius*, the RNase J orthologue Saci-aCPSF2 is a known exonuclease that digests 5′-p-RNAs while retaining some activity against 5′-ppp-RNAs[39,40]. We found that NAD–RNAs are efficiently degraded by *S. acidocaldarius* MW001 (WT)

S30 cell extracts and not by ΔSaci-aCPSF2 (KO) extract. Moreover, the NAD/ADPR–RNA ratio analysis for both WT and KO evidenced a ~4.5-fold increase in NAD–RNAs in the latter strain. Nonetheless, additional mechanistic studies are required to elucidate if Saci-aCPSF2 can release intact NAD molecules, as described for DXO/Rai1 homologs[3].

It is worth noting that, in hyperthermophilic environments, such as the natural habitats of *S. acidocaldarius*, NAD is quickly degraded into ADPR and Nm[14,15]. Therefore, hyperthermophilic organisms must have robust mechanisms to prevent the accumulation of toxic compounds generated via the thermal degradation of NAD. Here, we demonstrate that the half-life of NAD bound to RNA is 31 min at 85 °C, reinforcing the necessity of salvage pathways as described for free NAD[14,36].

Analysis of the ΔSaci_NudT5 strain evidenced a significant reduction in the amount of NAD and ADPR–RNAs. We hypothesize that the knockout of Saci_NudT5 directly impacts the NAD salvage pathway of *S. acidocaldarius*, which can lead to lower intracellular concentrations of this metabolite. Consequently, the process of NAD-capping might be suppressed. Moreover, if ADPR–RNAs are generated post-transcriptionally, either by an unknown NADase or by heat-degradation of NAD, the initial concentration of NAD–RNAs is a critical determinant of this process.

The identification of Saci_NudT5 as an ADPR-decapping enzyme suggests a scenario where NAD–RNAs are either i) directly degraded by Saci-aCPSF2 prior to decapping or ii) converted to ADPR–RNAs by non-characterized NADases or by thermal degradation, subsequently processed to 5′-p-RNAs by Saci_NudT5 and finally degraded by Saci-aCPSF2. Therefore, we propose that NAD-capping, ADPR-capping, and NAD metabolism are interconnected in *S. acidocaldarius* (Fig. 6).

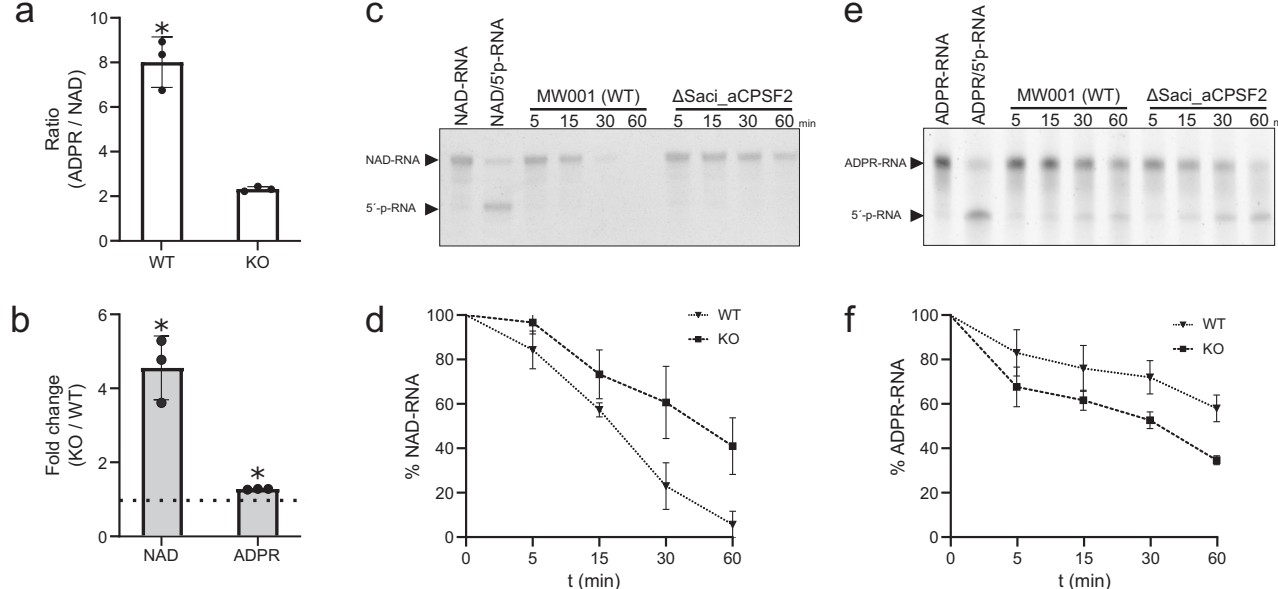

**Fig. 5 | Deletion of Saci-aCPSF2 impacts the levels of NAD–RNAs in *S. acid-ocaldarius*. a** Ratio of ADPR over NAD in nuclease P1 digested total RNA from *S. acidocaldarius* MW001 (WT) and ΔSaci-aCPSF2 (KO) determined by LC–MS/MS (Average of three independent experiments with error bars representing ±SD, two-sided unpaired *t* test *p = 0.0029). **b** Ratio of NAD and ADPR in nuclease P1 digested total RNA when compared between total RNA of the WT and KO strains as determined by LC–MS/MS (Average of three independent experiments with error bars representing ±SD; two-sided unpaired *t* test; NAD *p = 0.0002, ADPR *p = 0.0214). **c** Nuclease and decapping activity of S30 cell extracts (S30) from WT and KO using

in vitro transcribed NAD-Model RNA (38 nt). Reactions were incubated for up to 60 min at 65 °C, resolved on an APB polyacrylamide gels, and imaged. No NAD decapping was detected and **d** the NAD–RNA decay was quantified (Average of three independent experiments with error bars representing ±SD). **e** Parallel S30 cell extract incubations were performed with in vitro transcribed ADPR-Model RNA (38 nt) substrate. Conversion of ADPR–RNAs to 5′-p-RNA was identified and **f** ADPR–RNA decay was quantified (Average of three independent experiments with error bars representing ±SD). Source data are provided as a Source Data file.

## Methods

### Strains, plasmids, and oligonucleotides

All strains, plasmids, and oligonucleotide sequences used in this study are described in Supplementary Table 3. This work utilized *S. acidocaldarius* DSM639 MW001 (*S. acidocaldarius* MW001 for short), a uracil-auxotroph strain[44]. Cultures were grown aerobically at 120 rpm and 75 °C in Brock medium, pH 3.5[45]. The medium was supplied with 0.1% (w/v) NZ-Amine and 0.2% (w/v) dextrin, and 10 μg/ml uracil. *H. volcanii* was grown as previously described[26]. The remaining *E. coli* strains were grown aerobically at 180 rpm and 37 °C in LB medium (0.5% (w/v) yeast extract, 1% (w/v) tryptone, 1% (w/v) NaCl). For solid medium, LB medium was mixed with 1.5% (w/v) agar-agar and supplied with the respective antibiotic (0.001% (v/v)). Cell growth was achieved by monitoring the optical density of the cultures at 600 nm.

### Generation of a *S. acidocaldarius* ΔSaci-aCPSF2 strain

A markerless Saci-aCPSF2 deletion strain was generated using the deletion plasmid pSVA431[44]. The primers used for all steps are listed in Supplementary Table 3. With this method, the gene encoding Saci-aCPSF2 (SACI_RS11425) was deleted from the genome of *S. acidocaldarius* MW001 without interrupting its neighboring genes. The successful removal of the Saci-aCPSF2 gene was subsequently verified by Sanger sequencing of PCR products amplified from the deletion loci. For PCR analysis of *S. acidocaldarius* cells, 20 μl cell culture was lysed in 20 μl 0.2 M NaOH for 5 min at RT, neutralized by the addition of 80 μl 0.2 M Tris-HCl and 5 μl suspension was used in a 20 μl reaction using DreamTaq DNA Polymerase (Thermo Scientific). The genomic DNA of *S. acidocaldarius* was isolated from 2 ml late logarithmic-phase cultures using the NucleoSpin® Tissue Kit (Macherey-Nagel), according to the manufacturer's instructions for cultured cells.

### Preparation of S30 cell extracts for *S. acidocaldarius* MW001 and ΔSaci-aCPSF2

S30 cell extracts for *S. acidocaldarius* MW001 and MW001ΔSaci-aCPSF2 were obtained as previously described[46]. Briefly, cultures were grown in Brock media at 75 °C until OD600 of 0.4. 200 ml of culture were pelleted by centrifugation at 8,000 × *g* for 10 min at 4 °C and resuspended in lysis buffer (20 mM Tris/HCl pH = 7.4; 10 mM Mg-acetate; 50 mM NH4Cl; 1 mM DTT). Cells were lysed by adding lysozyme (30 min on ice) and sonication (5 min, 30 s cycles). Finally, extracts were centrifuged at 30,000 × *g* for 60 min, and the supernatant was collected and stored at −80 °C. The protein concentration was measured using a NanoPhotometer® NP80 (Implen).

### NAD- and ADPR–RNA decapping and degradation assays using S30 cell extracts

Briefly, 100 ng of 5′-NAD–RNA or 5′-ADPR–RNA were incubated with 0.25 μg of total protein from S30 cell extracts in reaction buffer (10 mM MgCl2, 20 mM Tris pH 6.5, and 10 mM KCl) at 65 °C for 5, 15, 30 and 60 min. The reaction was terminated by adding 5 μl of 2× APB-loading buffer (8 M Urea, 10 mM Tris-HCl pH 8, 50 mM EDTA, bromophenol blue, and xylene cyanol blue), and the samples were resolved on a 6% PAA, 0.2% APB, 1× TAE, 8 M urea gel. The gel was stained with SYBR™ Gold Nucleic Acid Gel Stain (Thermo Fisher), scanned with a Typhoon Trio (GE Healthcare), and analyzed with ImageJ (v. 1.54d).

### RNA extraction and quality control

*S. acidocaldarius* and *H. volcanii* cells were harvested during the mid-logarithmic growth phase. A 15 ml pellet was lysed with Trizol reagent (Thermo Fisher), and total RNA was extracted following the manufacturer's instructions. When needed, total RNA was treated with DNase I (NEB) according to the manufacturer's instruction and further purified using a Monarch® RNA Cleanup Kit (50 μg) (NEB). RNA

**Fig. 6 | Proposed model for the relationship between NAD metabolism and RNA turnover in *S. acidocaldarius*.** The thermal degradation of free NAD yields ADPR and Nm. In *S. acidocaldarius*, the NAD salvage pathway is proposed to recover NAD from Nm. ADPR is converted to Ribose−5-Phosphate (R5P) and AMP by Saci_NudT5. These products can be utilized by the pentose phosphate pathway or for ATP synthesis. NAD molecules covalently linked to RNA remain heat labile[14] at elevated temperatures. NAD−RNAs converted to ADPR−RNAs via thermal degradation are processed by Saci_NudT5, which releases Nm and 5′-p-RNA. A 5′−3′ exonuclease (Saci-aCPSF2) can potentially participate in the degradation of NAD−RNAs, either by initiating exonucleolytic attack prior to decapping or by degrading 5′-p-RNAs generated by Saci_NudT5.

integrity was monitored with agarose gels, and RNA concentrations were obtained either with an Implen NanoPhotometer® or with Qubit™ HS RNA assay kit, following the manufacturer's instructions.

### In vitro transcription of NAD− and ADPR−RNAs
Briefly, each 100 µl IVT reaction contained: 1 µM DNA template (Supplementary Table 3), 10 µl T7 RNA polymerase (50,000 U/ml), 1 mM of each GTP, UTP, CTP, and 4 mM of either NAD or ADPR. The reactions were performed in transcription buffer (40 mM Hepes/KOH pH 8, 22 mM MgCl$_2$, 5 mM DTT) and incubated for 2 h at 37 °C. The transcripts were purified using the Monarch® RNA Cleanup Kit (50 µg) (NEB) and verified on a 6% PAA, 1× TAE, 0.2% APB, and 8 M urea gel.

### Monitoring of NAD−RNA conversion to ADPR−RNA after heat treatment
Briefly, 50 ng of in vitro transcribed NAD−RNA was incubated at 75 °C or 85 °C for up to 2 hours in 50 mM Tris-HCl buffer (pH 6.5 at 75 °C or 85 °C). Aliquots were taken after 5, 15, 30, 60, and 120 min and used as substrates for ADPR-decapping assays with hNudT5, as described above. The reaction was terminated by adding 5 µl of 2× APB-loading buffer, and the samples resolved on a 6% PAA, 0.2% APB, 1× TAE, and 8 M urea gel. The gel was stained with SYBR™ Gold Nucleic Acid Gel Stain (Thermo Fisher) and visualized with an InstaS GelStick Imager (InstaS Science Imaging™).

To calculate the half-life of NAD covalently linked to RNA, 1 µg of in vitro transcribed NAD−RNA was heated at 85 °C for up to 2 h in 50 mM Tris-HCl buffer (pH 6.5 at 85 °C). Aliquots were taken after 5, 15, 30, 60, and 120 min and submitted to nuclease P1 digestion followed by LC−MS/MS analysis as described above. The calculated concentration of NAD in each time point was used to calculate the half-life of NAD as previously described[14].

### Relative quantification of NAD in total RNA from the model archaea *S. acidocaldarius* and *H. volcanii* by LC−MS/MS
In total, 600 µg of total RNA from either *S. acidocaldarius* or *H. volcanii* were divided into six 1.5 ml tubes, 100 µg per tube, and digested with either 10 U of nuclease P1 (NEB) or 10 U of heat-inactivated nuclease P1 for 1 h at 37 °C in a reaction volume of 100 µl.

NAD was quantified using a targeted multiple reaction monitoring (MRM) approach in negative ionization mode after chromatographic separation by reversed-phase chromatography according to the following method. The chromatographic separation was performed on an Agilent Infinity II 1260 HPLC system using a YMC C18 column (250 mm, 4.6 mm ID, YMC, Germany) at a constant flow rate of 0.8 ml/min and a constant temperature of 22 °C with mobile phase A being 0.4 % acetic acid (Sigma-Aldrich, USA) in water and phase B being 20% Methanol (Honeywell, Morristown, New Jersey, USA) in water. The injection volume was 100 µl. The mobile phase profile consisted of the following steps and linear gradients: 0−2 min constant at 0% B; 2−16 min from 0 to 35% B; 16−13 min from 35 to 100% B; 16−20 min constant at 100% B; 20−22 min from 100 to 0% B; 22−28 min constant at 0% B. An Agilent 6470 mass spectrometer was used in negative mode with an electrospray ionization source and the following conditions: ESI spray voltage 3500 V, sheath gas 400 °C at

11 l/min, nebulizer pressure 45 psi and drying gas 170 °C at 5 l/min. NAD was identified based on its specific mass transitions (662 (m/z) → 540 (m/z) and 662 (m/z) → 273 (m/z)) and retention time compared to standards. Extracted ion chromatograms of the compound-specific mass transitions were integrated using MassHunter software (Agilent, Santa Clara, CA, USA).

## Quantification of NAD, Nm, and ADPR isolated from nuclease P1 digested RNAs

Quantitative determination of NAD, Nm, and ADPR was performed using a targeted multiple reaction monitoring (MRM) approach in ion switching mode (NAD(+), Nm(+), ADPR(−)) after chromatographic separation by hydrophilic interaction chromatography according to the following method. The chromatographic separation was performed on an Agilent Infinity II 1290 HPLC system using a SeQuant ZIC-pHILIC column (150 × 2.1 mm, 5 μm particle size, peek coated, Merck) connected to a guard column of similar specificity (20 × 2.1 mm, 5 μm particle size, Phenomoenex) a constant flow rate of 0.1 ml/min and a constant temperature of 25° with mobile phase A being 10 mM ammonium acetate in water, pH 9, supplemented with medronic acid to a final concentration of 5 μM and phase B being 10 mM ammonium acetate in 90:10 acetonitrile to water, pH 9, supplemented with medronic acid to a final concentration of 5 μM.

The injection volume was 2 μl. The mobile phase profile consisted of the following steps and linear gradients: 0–1 min constant at 75% B; 1–6 min from 75 to 40% B; 6–9 min constant at 40% B; 9–9.1 min from 40 to 75% B; 9.1–20 min constant at 75% B. An Agilent 6495 ion funnel mass spectrometer was used in switch ionization mode with an electrospray ionization source and the following conditions: ESI spray voltage 3500 V, nozzle voltage 1000 V, sheath gas 400 °C at 9 l/min, nebulizer pressure 20 psig and drying gas 100 °C at 11 l/min. Compounds were identified based on their specific mass transition and retention time compared to standards, with NAD 664.1→542 and 664.1→428, Nm 123→80 and 123→78, and ADPR 558.1→345.9 and 558.1→78.9. Chromatograms were integrated using MassHunter software (Agilent, Santa Clara, CA, USA). Absolute concentrations were calculated using an external calibration curve prepared in the sample matrix. Mass transitions, collision energies, cell accelerator voltages, and dwell times have been optimized using chemically pure standards.

## Purification of recombinant NUDIX proteins

The genes encoding the Nudix proteins SACI_RS00730, Saci_NudT5, and SACI_RS00575 were cloned downstream of the 6× His-tag sequence on the vector pRSFDuet-1 using the restriction sites BamHI and HindIII. NUDIX domain mutant plasmids were generated by performing triple nucleotide exchange via site-directed mutagenesis on the plasmids. The thus generated plasmids were transformed into the expression strain *E. coli* Rosetta 2 DE3 pLysS (Novagen Darmstadt). Cells were grown in 1 L LB medium supplied with 30 μg/ml kanamycin at 37 °C, 200 rpm, and protein expression was induced at $OD_{600nm} = 0.6–0.8$ with 1 mM IPTG (for SACI_RS00730 and Saci_NudT5) or 0.1 mM IPTG for SACI_RS00575. After further incubation for 3–4 h at 37 °C, 200 rpm (for SACI_RS00730 and Saci_NudT5), or overnight at 18 °C, 200 rpm for SACI_RS00575, cells were harvested by centrifugation for 15 min at 12,000×g, 4 °C. Pellets were resuspended in 5 ml/g Wash Buffer (WB) (50 mM Tris-HCl, 1 M NaCl, 20 mM Imidazole, 10 mM $MgCl_2$, 1 mM DTT, 10% glycerol, pH 8.0), 1.5 mg lysozyme per gram cells was added to the suspension and incubated on ice for 30 min. Next, cells were cracked by sonication, and the supernatant was cleared by centrifugation (20 min at 30,000×g, room temperature). Subsequently, the lysate was incubated for 15 min at 75 °C, 500 rpm, to denature *E. coli* proteins. After another centrifugation step of 15 min at 16,000×g, 4 °C, the lysate was filtered using a Millex syringe filter (pore size 0.45 μm). One Pierce Centrifuge Column per protein was prepared by washing with several column volumes (cvs) of

20% ethanol and loaded with Ni-NTA Agarose (Qiagen) (stored in 20% ethanol) until each column was filled with ~2 ml resin. The columns were washed with 10 cvs double-distilled $H_2O$ (dd$H_2O$) followed by 10 cvs Wash Buffer. The lysate was loaded into a resin-filled column, and the flowthrough was saved. Columns were washed with 10 cvs Wash Buffer to remove nonspecifically bound proteins. For elution of the His-tagged proteins, the columns were subsequently washed with two times 1 cv of Elution Buffer 1 (WB with 100 mM Imidazole), four times 1 cv of Elution Buffer 2 (WB with 250 mM Imidazole), and three times 1 cv of Elution Buffer 3 (WB with 500 mM Imidazole). Protein elution fractions were analyzed via SDS-PAGE. Protein concentration was analyzed using a Qubit™ 2.0 Fluorometer and the Qubit™ Protein Assay Kit (ThermoFisher Scientific). Finally, proteins were stored at 4 °C.

In vitro protein expression was conducted using the NEBExpress® Cell-free *E. coli* Protein Synthesis System (NEB) according to the manufacturer's instructions. The plasmids carrying the N-terminally 6× His-tagged genes for SACI_RS02625 and its Nudix domain mutant, which were used as templates for the cell-free expression, were cloned by Genscript Inc. Subsequent purification of the in vitro produced proteins was performed using the NEBExpress® Ni Spin Columns (NEB) according to the manufacturer's protocol.

The pET28a-hNudT5 plasmid was transformed into the *E. coli* strain BL21 (DE3). The transformed cells were grown in LB media at 37 °C in the presence of 30 μg/mL kanamycin until $OD_{600}$ reached 0.8. *E. coli* BL21 (DE3) cells were then induced with 0.5 M IPTG, harvested after 3 h, and lysed by sonication (30 s, 50% power, five times) in HisTrap buffer A (25 mM Tris/HCl pH 8.0, 150 mM NaCl, 5 mM imidazole, 1 mM DTT). The lysate was clarified by centrifugation (16,000×g, 30 min, 4 °C), and the supernatant was applied to a Ni-NTA HisTrap column (GE Healthcare). The His-tagged protein was eluted with a gradient of HisTrap buffer B (HisTrap buffer A with 500 mM imidazole) and analyzed by SDS-PAGE. Subsequently, enzymes were purified via size exclusion chromatography with a Superose™ 6 Increase 30/100 GL column in gel filtration buffer (25 mM Tris/HCl pH 8.0, 150 mM NaCl). All purified protein samples were 95% pure, judging from SDS-PAGE.

## Evaluation of the NAD decapping and ADPR-decapping activity of NUDIX candidate proteins

The NUDIX candidates from *S. acidocaldarius*, NudC (NEB), and hNudT5 were used for decapping assays with NAD– and ADPR–RNAs. Briefly, for each reaction, 1 μl of RNA substrate (15 pmol), 0.5 μl of 10× NEBuffer r3.1, 2.5 μl Nuclease-Free $H_2O$ and 1 μl of the respective enzyme (15 pmol) were incubated at either 65 °C (for the *S. acidocaldarius* NUDIX) or 37 °C (for NudC and hNudT5) for 5 min. For each sample, a no-enzyme control was established. The reaction was terminated by adding 5 μl of 2× APB-loading buffer (8 M Urea, 10 mM Tris-HCl pH 8, 50 mM EDTA, bromophenol blue, and xylene cyanol blue), and the samples resolved on a 6% PAA, 0.2% APB, 1× TAE, 8 M urea gel. The gel was stained with SYBR™ Gold Nucleic Acid Gel Stain (Thermo Fisher) and visualized with an InstaS GelStick Imager (InstaS Science Imaging™).

## NAD captureSeq library preparation, sequencing, and data analysis

Briefly, as previously described, 600 μg of DNA-free total RNA from each organism was used as input for the preparation of NAD captureSeq libraries[1]. Each library was prepared in triplicates (ADPR+ A, B, C, and ADPRC− A, B, and C). Next, PCR products ranging from 150 bp to 300 bp were purified by Bluepippin size selection. The removal of primer dimers was evaluated using the Agilent DNA 1000 Kit (Agilent) on a Bioanalyzer 2100. The multiplexed library was submitted to NGS on an Illumina HiSeq 3000 or an Illumina MiniSeq in single-end mode and 150 nt read length.

Starting Gs of the raw reads and the 3′-adapter were trimmed using Cutadapt (v2.8) and quality checked with FASTQC (v0.11.9)[47,48]. Processed reads (≥18 nt) were mapped to the reference genome of either *S. acidocaldarius* or *H. volcanii* using Hisat2 (v2.2.1)[49]. After the strand-specific screening, HTSeq (v2.0.2) was used to count gene hits[50]. Statistical and enrichment analyses were performed with DESeq2 (v1.36.0)[20]. The Integrative Genomics Viewer (IGV, v2.13.2) was used to inspect and visualize candidate sequences[51]. Coding genes were clustered according to their respective arCOGs[29].

### dRNA-seq library preparation, sequencing, and data analysis
To identify transcription start sites (5′-ppp-RNA) in *S. acidocaldarius*, we applied the dRNA-seq technique[52]. Briefly, 5 µg of DNA-free total RNA was split into two tubes. One was treated with Terminator™ 5′-Phosphate-Dependent Exonuclease (TEX) (Lucigen, Epicenter) following the manufacturer's instructions, and the other was submitted to the same reaction but without enzyme. After digestion, the RNA was purified with Monarch® RNA Cleanup Kit (10 µg) (NEB), following the manufacturer's instructions. Illumina-compatible libraries were prepared using the NEBNext® Small RNA Library Prep Set for Illumina® (NEB). PCR size selection and quality control were performed as described above. The multiplexed library was submitted to NGS using single-end reads, 150 nt read-length on a HiSeq 3000.

For *H. volcanii*, previously published data were downloaded from the SRA database (PRJNA324298) and reanalyzed[26]. Raw reads were processed as described above. Transcription Start sites were detected using TSSAR: Transcription Start Site Annotation Regime Web Service (v1457945232)[53]. Primary transcription start sites matched with NAD−TSS were manually curated using Integrative Genomics Viewer (IGV)[51].

### sRNA-seq and ADPRC+ library comparison
Previously published sRNA-seq[23] data was downloaded from the SRA database (SRX2548838) and reanalyzed. Raw reads were processed as described above. After the strand-specific screening, HTSeq was used to count gene hits[50]. Next, genes were ranked according to their fractional representation in each dataset. The top 50 most abundant genes for each library were compared.

### Analysis of NAD−RNAs in total RNA with ADPRC-catalyzed biotinylation followed by qRT-PCR
Biotinylation of NAD−RNAs was performed as previously described[22] with the following adjustments. Total RNA was extracted with TriZol reagent (ThermoFischer) as described above. Total RNA (100 µg) was spiked-in with 100 ng of NAD-Model-RNA and incubated with 100 mM HEEB (N-[2-(2-hydroxyethoxy) ethyl]-biotinamide, CAS: 717119-80-7, 5 mM, Amatek scientific, catalog: B-1328) (1 M stock in DMSO), ADPRC (25 µg/ml, Sigma-Aldrich, catalog: A9106), and 0.4 U/µl of RNase Inhibitor (Takara Bio, catalog: 2313B) in 100 µl of ADPRC reaction buffer at 37 °C for 1 h (ADPRC+). Negative control was treated similarly but without ADPRC (ADPRC−). Next, the reaction was purified with phenol-chloroform and precipitated with ethanol. The RNA pellet was resuspended in 100 µl of nuclease-free water, incubated with streptavidin bead particles (MedChem Express) and 0.4 U/µl of Recombinant RNase Inhibitor (Takara Bio) at 25 °C for 30 min. Beads were washed four times with wash buffer (50 mM Tris-HCl pH 7.4 and 8 M urea), followed by three washes with nuclease-free water. The retained RNA was extracted with phenol-chloroform, ethanol precipitated, and resuspended in 12 µl of nuclease-free water. Reverse transcription was performed with random hexamers using SuperScript IV (Thermo-Fisher) following the manufacturer's instructions. qPCR was performed in triplicates using iQ™ SYBR® Green Supermix (Bio-Rad) to detect NAD−RNA or ppp-RNA. The ΔΔCt method was used to calculate the fold change enrichment (ADPRC+/ADPRC−), and Student's *t*-test was used to assess significance.

### Validation of NGS results with qPCR
Quantitative PCR (qPCR) was performed as described earlier[1] to validate the RNA enrichment observed in the NGS data on the cDNA level. Briefly, reactions were performed on a 20 µl scale in duplicate with 3 µl cDNA (1:50 diluted) as a template. qPCR was performed in a CFX Connect real-time PCR (BioRad) using the Brilliant III Ultra-Fast SYBRGreen qPCR Mastermix (Agilent). The data were collected and analyzed with the CFX Manager (v3.0.1224.1015) software. Millipore water was used as a negative control, and tRNA-Ile as an internal control gene. The $2^{-\Delta\Delta CT}$-method[54] was used to compare APDRC+ sample cDNA with the ADPRC− control cDNA. The primers used for qPCR analysis are listed in Supplementary Table 3.

### Protein sequence analysis for the discovery of potential NADases
To identify potential NADases in *S. acidocaldarius*, an HMMER (http://hmmer.org/) analysis using Pfam seed alignments (Downloaded in April 2023) (Supplementary Data 7) was performed. Briefly, the script hmmbuild was used to construct HMM profiles for the seed alignments of interest. Next, hmmsearch was used to screen all annotated proteins from *S. acidocaldarius*. HMMER results were filtered using a score ≥15.

### Reporting summary
Further information on research design is available in the Nature Portfolio Reporting Summary linked to this article.

## Data availability
The generated NGS data are stored at the European Nucleotide Archive (ENA) under the project number PRJEB48624. Source data for figures and supplementary figures are provided as a Source Data File. Source data are provided with this paper.

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

## Acknowledgements

We thank Anita Marchfelder for providing *H. volcanii* strains and growth protocols, Sonja-Verena Albers for assistance with *S. acidocaldarius* cultivation and manipulation, Jennifer Kothe for establishing *H. volcanii* growth in our laboratory, Peter Claus for the support during LC–MS/MS

experiments, Bruno Huettel for assistance with Illumina sequencing, and Julia Wiegel for technical support. This work was funded by the Deutsche Forschungsgemeinschaft (Heisenberg program to L.R., SPP 2330 project 464500427 and RTG 2355 to K.H.), the LOEWE Research Cluster: Diffusible Signals to L.R., the Max Planck Society (Max Planck Research Group Leader funding to K.H.) and from the European Research Council (ERC) under the European Union's Horizon 2020 research and innovation program (Grant agreement No. 882789 RNA-Coenzyme, to A.J.).

## Author contributions

J.V.G-F: Experimental design and data analysis. J.V.G-F and L.R.: conceptualization. R.B: *Sulfolobus acidocaldarius* Nudix proteins expression, purification, and generation of deletion mutants. H.G.MF. A.J., A.B., and J.S.: NAD captureSeq library preparation. N.Po. and K.H.: HNudT5 purification and assistance with APB-gels. N.Pa.: LC–MS/MS-mediated ADPR and NAD detection. J.V.G-F. and L.R. wrote the paper together with input from all authors.

## Funding

## Competing interests

The authors declare no competing interests.
