## [Peer Review File · Nature Communications]

Identification of NAD-RNA species and ADPR-RNA decapping in ArchaeaReviewer #1 (Remarks to the Author):

The manuscript "Identification of NAD-RNAs species and ADPR-RNA decapping in Archaea" describes identification of RNAs modified at their 5' end in two archaeobacterial species, *Sulfolobus acidocaldarius* and *Haloferax volcanii*. The authors first purified total RNA from these organisms and determined the levels of NAD-RNAs. In the case of *S. acidocaldarius* they detected cca 260 fmol per ug of RNA, the highest ever level detected in any organism. Next, using CaptureSeq they identified individual NAD-RNAs, about 80 in each species. Surprisingly, they then showed that the NAD cap does not protect RNA against degradation by a 5' to 3' exonuclease but rather it accelerates the degradation. Subsequently, NUDIX proteins were bioinformatically identified in *S. acidocaldarius*, expressed, and their ability to remove NAD caps from NAD-RNAs was tested. The experiments revealed that these candidate enzymes did not possess this de-NADing activity. Rather, one of them was able to remove ADPR from ADPR-RNA. The authors concluded that environments with high temperatures promote degradation of NAD into ADPR, which leads to formation of ADPR-RNA, and the need to remove this cap.

Overall, this is an interesting study, which includes some exciting findings, such as the increased susceptibility of NAD-RNAs to degradation by an archaeal 5' to 3' exonuclease, and the proposal for the presence of ADPR-RNAs in Archaea. However, experimental evidence for some conclusions is poor and some of the conclusions are debatable. Control experiments are needed to clarify the whole picture, as well as more thorough explanations of several aspects of the study. See comments for specific examples.

Major Comments

1/ Lines 112, 293 (line 323 in Methods): The authors used 2 ml of Trizol to extract RNA from 15 ml of mid-log growing cells (that is at least 15 OD units together). The recommended ratio is about 0.01-0.1 OD units per 1 ml of Trizol. From our experience, higher amounts of biological material affects efficiency of cell lysis and release of large RNA/rRNA from cell debris. The resulting total RNA, even if not degraded, contains a huge amount of small RNAs and is often characterized by "unexpected signal in the 5S region" by the Agilent Bioanalyser. The altered ratio of short RNAs: rRNAs substantially affects the capping quantification, which is normalised to total RNA. In other words, the "total" RNA used for quantitation of the presence of NAD in RNA may not be total RNA. Therefore, we recommend to use the NAD quantification data to demonstrate the presence of NAD on RNA but not to compare the overall level of Archaeal capping with other organisms – this information may be semiquantitative at best. At the very least, discuss this issue.

2/ What is the percentage of capping of individual RNAs? Does it match the relatively high abundance of NAD-RNAs as detected by the mass spectrometry from total RNA approach? Line 133: qPCR as described in the manuscript verifies only the quantification of the NAD capture protocol (using the same RNA), not the actual presence of the NAD caps in the identified RNAs. We suggest to use an independent approach, such as boronate polyacrylamide Northern blots, focusing on the most NAD-modified RNA(s) to provide a direct proof of the presence of this RNA modification.

3/ Lines 201, 289: Data supporting the statement that NAD-RNAs are preferential substrates for degradation by Saci-aCPSF2, an exonuclease, are poor (supplementary Figure 4). Size markers are missing and support for annotation of the bands is not clear. Include NudC treated samples and/or separate control lanes with ppp-RNA and NAD-RNA only (or at least ppp-RNA only and a mixture of ppp-RNA and NAD-RNA).

4/ Line 302: Can the presence of ADPR-RNAs be detected by mass spectrometry in total RNA to support the claims?

Minor Comments

5/ Line 167: The authors identified 41% of transcripts being capped at the start codon adenosine. This points to leaderless translation initiation. Provide some background information about leaderless translation. Does it have functional consequences? Can you speculate that the capped-

RNA leaderless translation may differ from the uncapped one?

6/ Line 186: An analysis of the possible alternative promoters of tRNAs and snoRNAs should be presented at least in supplementary data. Previously it was shown that up to 95 % of RNAs captured by the NAD capture seq method could be tRNAs even in the ADPRC minus control (see <https://doi.org/10.1016/j.celrep.2018.07.047>)

7/ Line 276: The claim that the manuscript shows that NAD capping of RNAs is common to all domains of life should be modified as NAD-capping of archaeal RNAs was already demonstrated in a study by Ruiz-Larrabeiti (2021).

8/ Lines 30, 267: For the general reader, explain why ADPR is toxic.

9/ Line 214: The reference should be to Fig. 3A (not 4A)

10/ Line 254: The pH of the Archaeal cytoplasm should be mentioned to be compared with the experimental conditions.

Reviewer #2 (Remarks to the Author):

NAD capped RNAs have been characterized in both prokaryotes and eukaryotes. In this study, Gomes-Filho and colleagues report the presence of the NAD capped RNAs in the archaeal domain. Here, they combined LC-MS and chemoenzymatic click chemistry approaches to quantify and profile NAD RNAs in two model archaea: *Sulfolobus acidocaldarius* and *Haloferax volcanii*. Their investigation of NAD transcription start-sites suggested RNAP ab initio capping as the primary mode of NAD capping in archaea. Unlike prokaryotes and eukaryotes that contain a battery of NAD cap decapping enzymes, their analysis revealed the absence of any NAD decapping activity in the archaeal NUDIX proteins, the homologs of which have been exhibited to show decapping activities in both prokaryote and eukaryotes. Instead, they identified Saci_NudT5 to exhibit activity against RppAp-RNA (ADPR-RNA) which can be formed (at least in vitro) from NAD capped RNAs following heat exposure to liberate the nicotinamide and generate ADPR-RNA. Overall, they claim that NAD capping influences the thermal stabilization of free NAD in *S. acidocaldarius* and other hyperthermophilic organisms. This is an interesting idea, but the manuscript is too preliminary to publication in Nature Communications as it stands and several fundamental issues that should be addressed to reinforce the author's claims are listed below.

Major Comments:

1. If NAD capped RNAs are thermally degraded into ADPR RNAs, authors should also present/assess the steady-state levels of ADPR RNAs in both *S. acidocaldarius* and *H. volcanii*. What is the concentration of free ADPR and ADPR capped RNAs in these cells, how does that compare to NAD/NAD RNA ratios and what is the ratio of NAD RNA to ADPR RNA?
2. The authors determined the concentrations of NAD to be 260 ± 72 fmol per μg RNA for *S. acidocaldarius* and 110 ± 9 fmol per μg RNA for *H. volcanii*. 260 fmol of NAD caps/ μg of total RNA. If one uses a conservative length of even 500nts as the average length of the transcript in archaea, this would suggest that around 4% of transcripts are NAD capped. Considering that ribosomal RNA makes ~90% of the total RNA, this would indicate that roughly all other transcripts are NAD capped. This would be very surprising and counter to the main thesis of the manuscript that NAD caps are converted to ADPR and degraded by Saci-aCPSF2. Although the proposed NAD capping as a strategy for the thermal stabilization of NAD in archaea is enticing, this has not been shown in the manuscript.
3. In Figure 4, the HNudT5 protein is used to decap RNAs, but a catalytically inactive mutant is missing. A mutant HNudT5 should also be included to confirm any activity is due to the protein and not a contaminant. In addition, why is an indirect assay used to determine whether the NAD RNA has been converted

to ADPR RNA? The question should be answered directly (possibly boronic acid PAGE can be used) and the concentrations of ADPR capped RNAs should be determined in the two organisms (as stated above).

It is also important to show that conversion of NAD RNA to ADPR RNA occurs in cells. Although this might occur in vitro, it's not a given that the same is true in cells. Also, since these organisms normally live at high temperatures, if the in vitro results applied to inside the cell, they would not be expected to have appreciable amounts of NAD or NAD RNA in cells.

4. In Supplementary Fig 4 a catalytically inactive mutant Saci-aCPSF2 should be included in these assays to confirm any activity is due to the protein and not a contaminant. Also in this figure, NAD- and ppp-RNAs are simultaneously used in the assay. Although the disappearance of each full length RNA is obvious, how do the authors know the identity of the decay fragments? They have specifically labeled the decay fragments as originating from NAD or ppp-RNA, but how do they know this?

NAD RNA and ppp RNA are shown to be degraded by Saci-aCPSF2 in this figure. Scai-aCPSF2 activity on monophosphate RNA should also be tested as a comparison especially since it is a homolog of RNaseJ a prominent monophosphate 5' end RNA nuclease.

5. Endogenous NAD capped transcripts were identified in the two organisms (86 NAD-RNAs in *H. volcanii* and 83 NAD-RNAs in *S. acidocaldarius*). A few of these should be validated with an independent approach. For example Northern blot of Boronic acid PAGE along with the appropriate controls.

6. Lastly, evidence that Saci-aCPSF2 can degrade NAD RNAs of HNudT5 can degrade ADPR RNA in cells is lacking. Genetic manipulations of knocking out the genes encoding Saci-aCPSF2 and HNudT5 should be carried out to ask whether these enzymes function on NAD or ADPR RNAs in cells and their consequence on the levels of these RNAs.

Minor Comments:

The authors should more carefully go through the manuscript for mistakes. A few examples are:

Line 50: Do the authors mean DXO/Rai1 instead of DXO/Rat1?

Line 68: To my knowledge, Rat1 has not been shown to be in mitochondria or function on mitochondrial RNA.

Line 214: The listing of "Fig 4A" does not appear to be correct. This figure does not show the 4 different *S. acidocaldarius* Nudix protein candidates as listed in the text.

Reviewer #3 (Remarks to the Author):

By NAD captureSeq, Gomes-Filho et al. have detected NAD caps on specific transcripts in the model archaea *Sulfolobus acidocaldarius* and *Haloferax volcanii* and deduced from the 5'-terminal RNA sequences that capping occurs by NAD incorporation during transcription initiation. They also report that in vitro these NAD caps slowly convert spontaneously to ADP ribose (ADPR) caps at high temperatures conducive to *S. acidocaldarius* growth and that the *S. acidocaldarius* protein Saci_NudT5, though unable to react with NAD caps, can remove ADPR caps from RNA. In addition, they present in vitro evidence that Saci-aCPSF2, the *S. acidocaldarius* ortholog of the bacterial 5'→3' exonuclease RNase J, can degrade NAD-capped RNA (NAD-RNA) faster than RNA bearing a 5' triphosphate (ppp-RNA). The authors propose a decay pathway for NAD-RNA in *S. acidocaldarius* involving spontaneous nicotinamide release, Saci_NudT5, and Saci-aCPSF2.

NAD-capped transcripts have previously been reported in bacterial and eukaryotic cells. This manuscript makes an important contribution to scientific knowledge by showing for the first time that NAD-capped transcripts are present in archaea, the third domain of life. A shortcoming of these studies is uncertainty as to the physiological significance of this discovery. In particular, it is not clear whether NAD capping affects the function or lifetime of RNA in *S. acidocaldarius* or whether the Saci_NudT5-dependent decapping pathway proposed in Figure 5 is fast enough to make a significant contribution to Saci-aCPSF2-mediated degradation of NAD-capped RNA in this organism. It remains possible that Saci-aCPSF2 degrades NAD-RNA faster than decapped RNA,

obviating the need for decapping as a prelude to RNA degradation. It's also possible that *S. acidocaldarius* contains an enzyme that accelerates decapping by catalyzing the release of nicotinamide from NAD caps so that the rate of decapping is not limited by the slow rate at which nicotinamide is released spontaneously. The impact of the authors' findings would be substantially enhanced by evidence addressing one or more of these questions.

Additional comments:

Lines 30-32 and 264-266. Presumably, the concentration of NAD far exceeds the concentration of NAD-RNA in *S. acidocaldarius*. If so, how can NAD-RNAs be thought to stabilize and store meaningful amounts of NAD, especially if this organism lacks an enzyme able to release NAD from these transcripts?

Lines 54-56. In *E. coli*, RNA with a 5'-monophosphate terminus is degraded by a cellular endonuclease, not 5'→3' exonucleases.

Figure 1. The retention time graphs in panels b and c should be labeled (active P1, inactivated P1) to make the differences easier to understand.

Figures 2 and 4 and Supplementary Figure 2. Many of the labels in these figures are illegibly small.

Supplementary Figure 4. This important figure merits inclusion as one of the main figures. Besides the NAD- and ppp-RNAs, it would also be very informative to know the relative reactivity of ADPR-RNA and p-RNA with Saci-aCPSF2. Examining these additional substrates would reveal whether or not decapping has the potential to accelerate RNA degradation by this enzyme. Finally, the figure legend or experimental procedures should describe the RNAs used as substrates in this experiment and explain why the degradation products of the capped and uncapped RNAs (mononucleotides?) don't co-migrate with one another.

Line 214. Figure 4A is mistakenly cited instead of Figure 3A.

Figure 3A. The meaning of the colors in this sequence alignment should be explained in the figure legend.

Lines 257-258. No direct evidence is presented to validate the inference that heating NAD-RNA generates ADPR-RNA. This presumably could be shown by digesting the RNA with nuclease P1 and then analyzing the products by mass spectrometry.

Lines 260-262 and Figure 4. It's not clear how half-lives for nicotinamide release could be calculated reliably from this data, as the reactions don't seem to conform to first-order kinetics. Instead, they appear to slow almost to a halt after 30 minutes. This is surprising for a hydrolytic reaction that is not enzyme-catalyzed since there is no catalyst with the potential to lose activity over time. The conformity or non-conformity of these reactions to first-order kinetics would be clearer if the data in panels (b) and (d) was plotted semilogarithmically instead of linearly.

Lines 721-722. The decay equation should be $dC/dt = -kC$, not $-k dC$.

Response to reviewer comments

Reviewer #1 (Remarks to the Author):

...Overall, this is an interesting study, which includes some exciting findings, such as the increased susceptibility of NAD-RNAs to degradation by an archaeal 5' to 3' exonuclease, and the proposal for the presence of ADPR-RNAs in Archaea. However, experimental evidence for some conclusions is poor and some of the conclusions are debatable. Control experiments are needed to clarify the whole picture, as well as more thorough explanations of several aspects of the study. See comments for specific examples.

Major Comments

1/ Lines 112, 293 (line 323 in Methods): The authors used 2 ml of Trizol to extract RNA from 15 ml of mid-log growing cells (that is at least 15 OD units together). The recommended ratio is about 0.01-0.1 OD units per 1 ml of Trizol. From our experience, higher amounts of biological material affects efficiency of cell lysis and release of large RNA/rRNA from cell debris. The resulting total RNA, even if not degraded, contains a huge amount of small RNAs and is often characterized by “unexpected signal in the 5S region” by the Agilent Bioanalyser. The altered ratio of short RNAs: rRNAs substantially affects the capping quantification, which is normalised to total RNA. In other words, the “total” RNA used for quantitation of the presence of NAD in RNA may not be total RNA. Therefore, we recommend to use the NAD quantification data to demonstrate the presence of NAD on RNA but not to compare the overall level of Archaeal capping with other organisms – this information may be semiquantitative at best. At the very least, discuss this issue.

Response: We thank the reviewer for bringing up this point and agree that this data should only be used for semiquantitative analysis of the presence of NAD-RNAs. To further address this issue, we performed new LC-MS measurements incorporating the reviewer's suggestions and exclusively used our results to compare levels within our experimental set-up. The manuscript text was adjusted accordingly (Lines 97-102, Fig. 1b-c, Fig. 5a-b).

2/ What is the percentage of capping of individual RNAs? Does it match the relatively high abundance of NAD-RNAs as detected by the mass spectrometry from total RNA approach?

Response: We toned the statements of high abundance of NAD-RNAs down in response to the reviewer's first major comment. In addition, we validated NAD captureSeq hits by ADPRC catalyzed biotinylation followed by qRT-PCR as described in (Niu et al. 2023). The results were added to Supplementary Figure 1h and discussed in lines 121-126.

Line 133: qPCR as described in the manuscript verifies only the quantification of the NAD capture protocol (using the same RNA), not the actual presence of the NAD caps in the identified RNAs.

Response: The reviewer is correct, this set of qPCRs were used to validate the NGS results for the cDNA generated by the NAD captureSeq protocol. This part of the text was rewritten to avoid any additional confusion. (Lines 115-121, Supplementary Figure 1g)

We suggest to use an independent approach, such as boronate polyacrylamide Northern blots, focusing on the most NAD-modified RNA(s) to provide a direct proof of the presence of this RNA modification.

Response: As noted above, we performed additional experiments for validation and added the results to Supplementary Figure 1h and lines 121-126.

3/ Lines 201, 289: Data supporting the statement that NAD-RNAs are preferential substrates for degradation by Saci-aCPSF2, an exonuclease, are poor (supplementary Figure 4). Size markers are missing and support for annotation of the bands is not clear. Include NudC treated samples and/or separate control lanes with ppp-RNA and NAD-RNA only (or at least ppp-RNA only and a mixture of ppp-RNA and NAD-RNA).

Response: The purification of Saci-aCPSF2 requires a denaturation step to release the protein from inclusion bodies, which incidentally leads to the need for refolding (Märtens et al. 2013). We designed and purified four putative catalytic inactivated mutants of Saci-aCPSF2 that showed high instability after refolding, with intense protein precipitation after heat-treatment (data not shown). Therefore, they would not be suitable controls for our nuclease assays that are performed at 65°C. We should note that all published Saci-aCPSF2 nuclease assays did not include a negative mutant, indicating the presence of folding or other issues for this protein. We then decided to remove the supplemental figure with *in vitro* Saci-aCPSF2 assays from our manuscript. However, we wished to test the effects of Saci-aCPSF2 in a more efficient way that considers possible unknown Saci-aCPSF2 interactions and unknown factors. We created a clean knockout strain (Δ Saci-aCPSF2) of Saci-aCPSF2 and analyzed exonuclease and decapping activities with S30 cell extracts that either contain or not contain Saci-aCPSF2. The manuscript was rewritten accordingly and includes the following novel exciting results:

First, total RNA from *S. acidocaldarius* MW001 (WT) and Δ Saci-aCPSF2 (KO) were probed for the presence of NAD- and ADPR-RNAs. LC-MS measurements evidenced a 4.3 ± 0.7 -fold increase (KO/WT) on the concentration of NAD-RNAs and a 1.25-fold increase (KO/WT) on the concentration of ADPR-RNAs (Fig. 5a-b).

Next, using S30 cell-extracts of both the WT and the KO, we evaluated the degradation of NAD- and ADPR-RNAs (Fig. 5c-f). Here, we visualized that NAD-RNAs are more stable when incubated with the S30 cell-extract from the KO in contrast to the WT. Furthermore, incubation of ADPR-RNA with S30 cell-extracts from the KO and WT evidenced the conversion to 5'-p-RNA, reinforcing the ADPR-

decapping capabilities of this organism. We believe that, put together, these data substantiate the relationship between Saci-aCPSF2 and NAD-RNA turnover. (Lines 263-280, Fig. 5a-f)

4/ Line 302: Can the presence of ADPR-RNAs be detected by mass spectrometry in total RNA to support the claims?

Response: We performed this experiment and succeeded in detecting ADPR-RNAs in both *S. acidocaldarius* WT and in the Δ Saci-aCPSF2 strain (Fig. 5a). This result highlights a novel archaeal RNA cap.

Minor Comments

5/ Line 167: The authors identified 41% of transcripts being capped at the start codon adenosine. This points to leaderless translation initiation. Provide some background information about leaderless translation.

Does it have functional consequences? Can you speculate that the capped-RNA leaderless translation may differ from the uncapped one?

Response: Leaderless transcripts are rather common in Haloarchaea and Sulfolobales, and their impact on translation has been discussed (Gelsinger et al. 2020). So far, the few characterized archaeal translation initiation factors seem to bind specifically to 5'-ppp-mRNAs (Arkhipova et al. 2015; Bassani et al. 2019). Thus, one could hypothesize that the presence of a NAD-cap might impair this process. Moreover, the binding of the translation initiation factor a/eIF2(- γ) to the 5'-end of mRNAs acts as a protection mechanism to counteract 5'->3' exonucleases (Hasenöhrl et al. 2008). Finally, if the NAD-cap prevents the binding of translation factors, not only translation would be compromised but also the degradation of NAD-RNAs by Saci-aCPSF2 would be favored in *S. acidocaldarius*. Future binding assays with *S. acidocaldarius* translation initiation factors to NAD- and ADPR-RNAs will be helpful to answer this question.

6/ Line 186: An analysis of the possible alternative promoters of tRNAs and snoRNAs should be presented at least in supplementary data. Previously it was shown that up to 95 % of RNAs captured by the NAD capture seq method could be tRNAs even in the ADPRC minus control (see <https://doi.org/10.1016/j.celrep.2018.07.047>)

Response: In relation to alternative promoters of tRNAs, we do not think that this is the case for *S. acidocaldarius* and *H. volcanii*. In both cases, when we analyse the presence of primary transcription start sites (pTSS) detected by our dRNA-seq analysis, we do not see a difference between pTSSs, and NAD transcription start sites (NAD-TSSs). This correlation strongly suggests that the same promoter is used for both canonical and non-canonical transcription initiation.

For C/D box sRNAs, eight out of eleven presents equivalent pTSS and NAD-TSS pointing to a similar scenario as described for tRNAs (Lines 180-184).

7/ Line 276: The claim that the manuscript shows that NAD capping of RNAs is common to all

domains of life should be modified as NAD-capping of archaeal RNAs was already demonstrated in a study by Ruiz-Larrabeiti (2021).

Response: To our knowledge, the indicated article is a preprint and has not been certified by peer review.

8/ Lines 30, 267: For the general reader, explain why ADPR is toxic.

Response: We modified the text accordingly (Lines 65-66).

9/ Line 214: The reference should be to Fig. 3A (not 4A)

Response: Figure numbering was corrected.

10/ Line 254: The pH of the Archaeal cytoplasm should be mentioned to be compared with the experimental conditions.

Response: The membrane barrier is especially important in the archaeal genus *Sulfolobus*, as these species maintain an almost neutral cytoplasmic pH of 6.5, albeit growing in an acidic environment with a pH of 2.0 to 4.0 (Brock et al. 1972). We modified the text accordingly (Lines 238-241).

Reviewer #2 (Remarks to the Author):

NAD capped RNAs have been characterized in both prokaryotes and eukaryotes. In this study, Gomes-Filho and colleagues report the presence of the NAD capped RNAs in the archaeal domain. Here, they combined LC-MS and chemoenzymatic click chemistry approaches to quantify and profile NAD RNAs in two model archaea: *Sulfolobus acidocaldarius* and *Haloferax volcanii*. Their investigation of NAD transcription start-sites suggested RNAP ab initio capping as the primary mode of NAD capping in archaea. Unlike prokaryotes and eukaryotes that contain a battery of NAD cap decapping enzymes, their analysis revealed the absence of any NAD decapping activity in the archaeal NUDIX proteins, the homologs of which have been exhibited to show decapping activities in both prokaryote and eukaryotes. Instead, they identified Saci_NudT5 to exhibit activity against RppAp-RNA (ADPR-RNA) which can be formed (at least in vitro) from NAD capped RNAs following heat exposure to liberate the nicotinamide and generate ADPR-RNA. Overall, they claim that NAD capping influences the thermal stabilization of free NAD in *S. acidocaldarius* and other hyperthermophilic organisms. This is an interesting idea, but the manuscript is too preliminary to publication in Nature Communications as it stands and several fundamental issues that should be addressed to reinforce the author's claims are listed below.

Major Comments:

1. If NAD capped RNAs are thermally degraded into ADPR RNAs, authors should also present/assess the steady-state levels of ADPR RNAs in both *S. acidocaldarius* and *H. volcanii*. What is the concentration of free ADPR and ADPR capped RNAs in these cells, how does that compare to NAD/NAD RNA ratios and what is the ratio of NAD RNA to ADPR RNA?

Response: We have added LC-MS data to show thermal degradation of NAD (new Fig. 4d), detected ADPR-RNA in *S. acidocaldarius* total RNA and analysed relative ADPR vs NAD-RNA levels (new Fig. 5a-b). Detailed metabolomics analyses would be necessary to quantify the steady-state levels of ADPR RNAs and require additional collaborations. We believe that the added data highlight the key observation of NAD-RNA degradation and ADPR-detection.

2. The authors determined the concentrations of NAD to be 260 ± 72 fmol per μg RNA for *S. acidocaldarius* and 110 ± 9 fmol per μg RNA for *H. volcanii*. 260 fmol of NAD caps/ μg of total RNA. If one uses a conservative length of even 500nts as the average length of the transcript in archaea, this would suggest that around 4% of transcripts are NAD capped. Considering that ribosomal RNA makes ~90% of the total RNA, this would indicate that roughly all other transcripts are NAD capped. This would be very surprising and counter to the main thesis of the manuscript that NAD caps are converted to ADPR and degraded by Saci-aCPSF2. Although the proposed NAD capping as a strategy for the thermal stabilization of NAD in archaea is enticing, this has not been shown in the manuscript.

Response: Thank you very much. Encouraged by the comments of reviewer 1, we decided to use our LC-MS only semi-quantitatively and to not establish comparisons with other organisms. Thus, all the calculations and ratios presented in the manuscript are based on results obtained inside our experimental set-up. We further modified the manuscript to better explain the proposed scenario of NAD-RNA degradation by Saci-aCPSF2 and heat (Fig. 4, lines 238-241, 248-253; Fig. 5a-d, lines 263-280).

3. In Figure 4, the HNudT5 protein is used to decap RNAs, but a catalytically inactive mutant is missing. A mutant HNudT5 should also be included to confirm any activity is due to the protein and not a contaminant.

Response: The enzyme human NudT5 (hNudT5) serves as a positive control (like NudC in the NAD-decapping assays) and in our experimental set-up, all *Sulfolobus* NUDIX proteins have their corresponding inactive mutants (NDM) which show no activity in any of the tested conditions. As our focus was the *S. acidocaldarius* NUDIX and the mechanism/activity of hNudT5 has been described elsewhere (Arimori et al. 2011; Zha et al. 2006) we believe that the creation of hNudT5 mutants is not the focus of our study.

In addition, why is an indirect assay used to determine whether the NAD RNA has been converted to ADPR RNA? The question should be answered directly (possibly boronic acid PAGE can be used) and the concentrations of ADPR capped RNAs should be determined in the two organisms (as stated above).

Response: We did use Boronic Acid Page to verify the conversion of NAD- to ADPR-RNA. The use of an enzyme was required because the migration of ADPR/NAD-RNA cannot be differentiated in APB-gels. The presence of cis-diol in both molecules (NAD and ADPR) results in a similar shift in the gel. Thus, we solved this issue by using NudT5 enzymes (either Human or from *Sulfolobus*), which specifically cleave ADPR-RNA and not NAD-RNA. In addition, we performed LC-MS analysis of nuclease P1 digested heat degraded NAD-RNAs (as suggested by Reviewer 3) and by tracking Nicotinamide, ADPR, and NAD we confirmed the conversion of NAD-RNAs to ADPR-RNAs. These results were added to Figure 4 and the text was modified accordingly (Lines 238-241, 248-253).

It is also important to show that conversion of NAD RNA to ADPR RNA occurs in cells. Although this might occur in vitro, it's not a given that the same is true in cells. Also, since these organisms normally live at high temperatures, if the in vitro results applied to inside the cell, they would not be expected to have appreciable amounts of NAD or NAD-RNA in cells.

Response: We performed new LC-MS analysis to detect NAD- and ADPR-RNAs and found an 8 ± 0.92 ratio of ADPR- to NAD-RNAs in *S. acidocaldarius* WT (Fig. 5a). Additionally, the knockout of Saci-aCPSF2 presents a ratio of 2.3 ± 0.1 , pointing to an accumulation of NAD-RNAs in this genetic background (Fig. 5a, Lines 263-280).

4. In Supplementary Fig 4 a catalytically inactive mutant Saci-aCPSF2 should be included in these

assays to confirm any activity is due to the protein and not a contaminant. Also in this figure, NAD- and ppp-RNAs are simultaneously used in the assay. Although the disappearance of each full length RNA is obvious, how do the authors know the identity of the decay fragments? They have specifically labeled the decay fragments as originating from NAD or ppp-RNA, but how do they know this? NAD RNA and ppp RNA are shown to be degraded by Saci-aCPSF2 in this figure. Scai-aCPSF2 activity on monophosphate RNA should also be tested as a comparison especially since it is a homolog of RNaseJ a prominent monophosphate 5' end RNA nuclease.

Response: The purification of Saci-aCPSF2 requires a denaturation step to release the protein from inclusion bodies, which incidentally leads to the need for refolding (Märtens et al. 2013). We designed and purified four putative catalytic inactivated mutants of Saci-aCPSF2 that showed high instability after refolding, with intense protein precipitation after heat-treatment (data not shown). Therefore, they would not be suitable controls for our nuclease assays that are performed at 65°C. We should note that all published Saci-aCPSF2 nuclease assays did not include a negative mutant, indicating the presence of folding or other issues for this protein. We then decided to remove the supplemental figure with *in vitro* Saci-aCPSF2 assays from our manuscript. However, we wished to test the effects of Saci-aCPSF2 in a more efficient way that considers possible unknown Saci-aCPSF2 interactions and unknown factors. We created a clean knockout strain (Δ Saci-aCPSF2) of Saci-aCPSF2 and analyzed exonuclease and decapping activities with S30 cell extracts that either contain or not contain Saci-aCPSF2. The manuscript was rewritten accordingly and includes the following novel exciting results:

First, total RNA from *S. acidocaldarius* MW001 (WT) and Δ Saci-aCPSF2 (KO) were probed for the presence of NAD- and ADPR-RNAs. LC-MS measurements evidenced a 4.3 ± 0.7 -fold increase (KO/WT) on the concentration of NAD-RNAs and a 1.25-fold increase (KO/WT) on the concentration of ADPR-RNAs (Figure 5a-b).

Next, using S30 cell-extracts of both the WT and the KO, we evaluated the degradation of NAD- and ADPR-RNAs (Figure 5c-f). Here, we visualized that NAD-RNAs are more stable when incubated with the S30 cell-extract from the KO in contrast to the WT. Furthermore, incubation of ADPR-RNA with S30 cell-extracts from the KO and WT evidenced the conversion to 5'-p-RNA, reinforcing the ADPR-decapping capabilities of this organism. We believe that, put together, these data substantiate the relationship between Saci-aCPSF2 and NAD-RNA turnover (Lines 263-280, Fig. 5a-f).

5. Endogenous NAD capped transcripts were identified in the two organisms (86 NAD-RNAs in *H. volcanii* and 83 NAD-RNAs in *S. acidocaldarius*). A few of these should be validated with an independent approach. For example Northern blot of Boronic acid PAGE along with the appropriate controls.

Response: We included an independent approach and validated NAD captureSeq hits by ADPRC catalyzed biotinylation of NAD-RNAs followed by qRT-PCR as described in (Niu et al. 2023) (Supplementary Figure 1h, lines 121-126).

6. Lastly, evidence that Saci-aCPSF2 can degrade NAD-RNAs or HNudT5 can degrade ADPR-RNA

in cells is lacking. Genetic manipulations of knocking out the genes encoding Saci-aCPSF2 and HNudT5 should be carried out to ask whether these enzymes function on NAD or ADPR-RNAs in cells and their consequence on the levels of these RNAs.

Response: Using S30 cell-extracts of both the wild-type and a Saci-aCPSF2 knockout strain, we evaluated the degradation of NAD- and ADPR-RNAs. Here, we visualized that NAD-RNAs are more stable when incubated with the S30 cell-extract from the KO in contrast to the WT (Figure 5c-d). Furthermore, incubation of ADPR-RNA with S30 cell-extracts from the KO and WT evidenced the conversion to 5'-p-RNA, reinforcing the ADPR-decapping capabilities of this organism (Figure 5e-f). We believe that, put together, these data point to the relationship between Saci-aCPSF2 and NAD-RNA turnover (Lines 263-280).

Minor Comments:

The authors should more carefully go through the manuscript for mistakes. A few examples are:

Line 50: Do the authors mean DXO/Rai1 instead of DXO/Rat1?

Line 68: To my knowledge, Rat1 has not been shown to be in mitochondria or function on mitochondrial RNA.

Line 214: The listing of "Fig 4A" does not appear to be correct. This figure does not show the 4 different *S. acidocaldarius* Nudix protein candidates as listed in the text.

Response: We thank the reviewer for pointing out these mistakes and corrected these in the revised version.

Reviewer #3 (Remarks to the Author):

By NAD captureSeq, Gomes-Filho et al. have detected NAD caps on specific transcripts in the model archaea *Sulfolobus acidocaldarius* and *Haloferax volcanii* and deduced from the 5'-terminal RNA sequences that capping occurs by NAD incorporation during transcription initiation. They also report that in vitro these NAD caps slowly convert spontaneously to ADP ribose (ADPR) caps at high temperatures conducive to *S. acidocaldarius* growth and that the *S. acidocaldarius* protein Saci_NudT5, though unable to react with NAD caps, can remove ADPR caps from RNA. In addition, they present in vitro evidence that Saci-aCPSF2, the *S. acidocaldarius* ortholog of the bacterial 5'→3' exonuclease RNase J, can degrade NAD-capped RNA (NAD-RNA) faster than RNA bearing a 5' triphosphate (ppp-RNA). The authors propose a decay pathway for NAD-RNA in *S. acidocaldarius* involving spontaneous nicotinamide release, Saci_NudT5, and Saci-aCPSF2.

NAD-capped transcripts have previously been reported in bacterial and eukaryotic cells. This manuscript makes an important contribution to scientific knowledge by showing for the first time that NAD-capped transcripts are present in archaea, the third domain of life.

A shortcoming of these studies is uncertainty as to the physiological significance of this discovery. In particular, it is not clear whether NAD capping affects the function or lifetime of RNA in *S. acidocaldarius* or whether the Saci_NudT5-dependent decapping pathway proposed in Figure 5 is fast enough to make a significant contribution to Saci-aCPSF2-mediated degradation of NAD-capped RNA in this organism. It remains possible that Saci-aCPSF2 degrades NAD-RNA faster than decapped RNA, obviating the need for decapping as a prelude to RNA degradation.

It's also possible that *S. acidocaldarius* contains an enzyme that accelerates decapping by catalyzing the release of nicotinamide from NAD caps so that the rate of decapping is not limited by the slow rate at which nicotinamide is released spontaneously.

Response: We thank the reviewer for this suggestion. We performed a HMMER analysis using seed alignments of Pfams associated to NADases (Supplementary File 3) and detected nine potential NAD consuming enzymes in *S. acidocaldarius* (Supplementary Table 6). A brief discussion was added to the manuscript (Supplementary File 3, Supplementary Table 6, and Lines 228-231).

The impact of the authors' findings would be substantially enhanced by evidence addressing one or more of these questions.

Additional comments:

Lines 30-32 and 264-266. Presumably, the concentration of NAD far exceeds the concentration of NAD-RNA in *S. acidocaldarius*. If so, how can NAD-RNAs be thought to stabilize and store meaningful amounts of NAD, especially if this organism lacks an enzyme able to release NAD from these transcripts?

Response: Previous studies demonstrated that the 5'→3' exonuclease Xrn1 is directly involved in the degradation of NAD-RNAs, releasing single nucleotides and NAD, and can be involved in maintaining mitochondrial NAD levels (Sharma et al. 2022). Our results (Figure 5a-d, Lines 263-280) suggest that Saci-aCPSF2 performs a similar role in *S. acidocaldarius*, at least in the degradation of NAD-RNAs. Future studies will be necessary to confirm the impacts on the concentrations of free NAD.

Lines 54-56. In *E. coli*, RNA with a 5'-monophosphate terminus is degraded by a cellular endonuclease, not 5'→3' exonucleases.

Response: We modified the text accordingly (line 49).

Figure 1. The retention time graphs in panels b and c should be labeled (active P1, inactivated P1) to make the differences easier to understand.

Response: This figure was remade.

Figures 2 and 4 and Supplementary Figure 2. Many of the labels in these figures are illegibly small.

Response: Supplementary Figure 2 is now Supplementary Figure 1, the suggestion of bigger labels as incorporated.

Supplementary Figure 4. This important figure merits inclusion as one of the main figures. Besides the NAD- and ppp-RNAs, it would also be very informative to know the relative reactivity of ADPR-RNA and p-RNA with Saci-aCPSF2. Examining these additional substrates would reveal whether or not decapping has the potential to accelerate RNA degradation by this enzyme. Finally, the figure legend or experimental procedures should describe the RNAs used as substrates in this experiment and explain why the degradation products of the capped and uncapped RNAs (mononucleotides?) don't co-migrate with one another.

Response: In accordance with the points raised by reviewer 1 and 2, we removed this supplemental figure and added a novel figure (Fig. 5) to address the processing of capped RNA. The purification of Saci-aCPSF2 requires a denaturation step to release the protein from inclusion bodies, which incidentally leads to the need for refolding (Märtens et al. 2013). We designed and purified four putative catalytic inactivated mutants of Saci-aCPSF2 that showed high instability after refolding, with intense protein precipitation after heat-treatment (data not shown). Therefore, they would not be suitable controls for our nuclease assays that are performed at 65°C. We should note that all published Saci-aCPSF2 nuclease assays did not include a negative mutant, indicating the presence of folding or other issues for this protein. We then decided to remove the supplemental figure with *in vitro* Saci-aCPSF2 assays from our manuscript. However, we wished to test the effects of Saci-aCPSF2 in a more efficient way that considers possible unknown Saci-aCPSF2 interactions and unknown factors. We created a clean knockout strain (Δ Saci-aCPSF2) of Saci-aCPSF2 and analyzed

exonuclease and decapping activities with S30 cell extracts that either contain or not contain Saci-aCPSF2. The manuscript was rewritten accordingly and includes the following novel exciting results: First, total RNA from *S. acidocaldarius* MW001 (WT) and Δ Saci-aCPSF2 (KO) were probed for the presence of NAD- and ADPR-RNAs. LC-MS measurements evidenced a 4.3 ± 0.7 -fold increase (KO/WT) on the concentration of NAD-RNAs and a 1.25-fold increase (KO/WT) on the concentration of ADPR-RNAs (Figure 5a-b).

Next, using S30 cell-extracts of both the WT and the KO, we evaluated the degradation of NAD- and ADPR-RNAs (Figure 5c-f). Here, we visualized that NAD-RNAs are more stable when incubated with the S30 cell-extract from the KO in contrast to the WT. Furthermore, incubation of ADPR-RNA with S30 cell-extracts from the KO and WT evidenced the conversion to 5'-p-RNA, reinforcing the ADPR-decapping capabilities of this organism. We believe that, put together, these data substantiate the relationship between Saci-aCPSF2 and NAD-RNA turnover. (Lines 263-280, Fig. 5a-f)

Line 214. Figure 4A is mistakenly cited instead of Figure 3A.

Response: The figures are properly referenced now.

Figure 3A. The meaning of the colors in this sequence alignment should be explained in the figure legend.

Response: We modified the legend accordingly.

Lines 257-258. No direct evidence is presented to validate the inference that heating NAD-RNA generates ADPR-RNA. This presumably could be shown by digesting the RNA with nuclease P1 and then analyzing the products by mass spectrometry.

Response: To better represent the half-life of NAD covalently linked to RNA, we performed the experiment as suggested by the reviewer. Briefly, LC-MS detection of Nicotinamide, ADPR, and NAD following heat degradation and nuclease P1 digestion of NAD-RNAs evidenced the conversion of NAD- to ADPR- RNAs. These results were added to Figure 4c and 4d and the text was modified accordingly (Lines 248-253).

Lines 260-262 and Figure 4. It's not clear how half-lives for nicotinamide release could be calculated reliably from this data, as the reactions don't seem to conform to first-order kinetics. Instead, they appear to slow almost to a halt after 30 minutes. This is surprising for a hydrolytic reaction that is not enzyme-catalyzed since there is no catalyst with the potential to lose activity over time. The conformity or non-conformity of these reactions to first-order kinetics would be clearer if the data in panels (b) and (d) was plotted semilogarithmically instead of linearly.

Response: We thank the reviewer for pointing this out. While the conversion from NAD-RNA to ADPR-RNA itself is not enzyme-catalyzed, the conversion from ADPR-RNA to 5'-p-RNA is. Since this could be the limiting factor, we incorporated the reviewer's suggestion and performed LC-MS analysis following heat treatment and nuclease P1 digestion of NAD-RNAs (Fig. 4d). The obtained results point to a clear first-order decay (Figure 4d). From this data, a NAD(-RNA) half-life of about 31

minutes points to a slightly longer half-life than free NAD (24 minutes) (Hachisuka et al. 2017) (Lines 248-253).

Lines 721-722. The decay equation should be $dC/dt = -kC$, not $-kdC$.

Response: We modified the text accordingly.

Bibliography

- Arimori, Takao; Tamaoki, Haruhiko; Nakamura, Teruya; Kamiya, Hiroyuki; Ikemizu, Shinji; Takagi, Yasumitsu et al. (2011): Diverse substrate recognition and hydrolysis mechanisms of human NUDT5. In *Nucleic Acids Research* 39 (20), pp. 8972–8983. DOI: 10.1093/nar/gkr575.
- Arkhipova, Valentina; Stolboushkina, Elena; Kravchenko, Olesya; Kljashtorny, Vladislav; Gabdulkhakov, Azat; Garber, Maria et al. (2015): Binding of the 5'-Triphosphate End of mRNA to the γ -Subunit of Translation Initiation Factor 2 of the Crenarchaeon *Sulfolobus solfataricus*. In *Journal of Molecular Biology* 427 (19), pp. 3086–3095. DOI: 10.1016/j.jmb.2015.07.020.
- Bassani, Flavia; Zink, Isabelle Anna; Pribasnig, Thomas; Wolfinger, Michael T.; Romagnoli, Alice; Resch, Armin et al. (2019): Indications for a moonlighting function of translation factor aIF5A in the crenarchaeum *Sulfolobus solfataricus*. In *RNA Biology* 16 (5), pp. 675–685. DOI: 10.1080/15476286.2019.1582953.
- Brock, T. D.; Brock, K. M.; Belly, R. T.; Weiss, R. L. (1972): *Sulfolobus*: a new genus of sulfur-oxidizing bacteria living at low pH and high temperature. In *Archiv fur Mikrobiologie* 84 (1), pp. 54–68. DOI: 10.1007/BF00408082.
- Gelsinger, Diego Rivera; Dallon, Emma; Reddy, Rahul; Mohammad, Fuad; Buskirk, Allen R.; DiRuggiero, Jocelyne (2020): Ribosome profiling in archaea reveals leaderless translation, novel translational initiation sites, and ribosome pausing at single codon resolution. In *Nucleic Acids Research* 48 (10), pp. 5201–5216. DOI: 10.1093/nar/gkaa304.
- Hachisuka, Shin-ichi; Sato, Takaaki; Atomi, Haruyuki (2017): Metabolism Dealing with Thermal Degradation of NAD⁺ in the Hyperthermophilic Archaeon *Thermococcus kodakarensis*. In *J Bacteriol* 199 (19). DOI: 10.1128/JB.00162-17.
- Hasenöhrl, David; Lombo, Tania; Kaberdin, Vladimir; Londei, Paola; Bläsi, Udo (2008): Translation initiation factor a/eIF2(- γ) counteracts 5' to 3' mRNA decay in the archaeon *Sulfolobus solfataricus*. In *Proc Natl Acad Sci USA* 105 (6), pp. 2146–2150. DOI: 10.1073/pnas.0708894105.
- Märtens, Birgit; Amman, Fabian; Manoharadas, Salim; Zeichen, Lukas; Orell, Alvaro; Albers, Sonja-Verena et al. (2013): Alterations of the transcriptome of *Sulfolobus acidocaldarius* by exoribonuclease aCPSF2. In *PLoS ONE* 8 (10), e76569. DOI: 10.1371/journal.pone.0076569.
- Niu, Kongyan; Zhang, Jinyang; Ge, Shuwen; Li, Dean; Sun, Kunfeng; You, Yingnan et al. (2023): ONE-seq: epitranscriptome and gene-specific profiling of NAD-capped RNA. In *Nucleic Acids Research* 51 (2), e12. DOI: 10.1093/nar/gkac1136.
- Sharma, Sunny; Yang, Jun; Grudzien-Nogalska, Ewa; Shivas, Jessica; Kwan, Kelvin Y.; Kiledjian, Megerditch (2022): Xrn1 is a deNADding enzyme modulating mitochondrial NAD-capped RNA. In *Nat Commun* 13 (1), p. 889. DOI: 10.1038/s41467-022-28555-7.
- Zha, Manwu; Zhong, Chen; Peng, Yingjie; Hu, Hongyu; Ding, Jianping (2006): Crystal structures of human NUDT5 reveal insights into the structural basis of the substrate specificity. In *Journal of Molecular Biology* 364 (5), pp. 1021–1033. DOI: 10.1016/j.jmb.2006.09.078.

Reviewer #1 (Remarks to the Author):

The revised version is much improved. Overall, the two most exciting findings are: (i) *S. acidocaldarius* contains also ADPR (as a result of thermal degradation) at 5' ends of some RNAs and (ii) ADPR is removed by a dedicated nudix protein, *Saci_NudT5*. However, several key experiments are still missing, namely a direct proof of NAD-capping of selected RNAs and effects of *Saci_NudT5* deletion/depletion on ADPR-capped RNAs. For details, see comments.

Comments:

1/ The authors have not confirmed the 5' end modifications of RNAs by NAD by a truly independent approach such as northern analysis using boronate gels. The RT-qPCR results are a good addition but still, they are based on an indirect approach. If you want to convince the field, perform northern blot analysis with boronate gels and RNA(s) that are abundant and NAD-capped. Alternatively (if it is the case), explain that the NAD-capped RNAs represent such a small fraction of the respective RNAs that it is not possible to detect them with boronate gels.

2/ Line 277: You say "ADPR-RNA was processed into 5'-p-RNA at a slightly faster rate in the KO S30 extract (Fig. 5e-f). Altogether, these results suggest that the 5'-3' exonuclease *Saci-aCPSF2* plays an essential role in the turnover of NAD-capped RNAs and reinforces the ADPR-decapping activity in *S. acidocaldarius*." If it reinforced ADPR decapping, the degradation rate of the *Saci-aCPSF2* deletion strain should be slower than wt, not faster. The data actually suggest the opposite than you claim. Discuss/explain. Also, to support the main story, it would be highly desirable to create a knock-out of *Saci_NudT5* (or at least a depletion strain) and test the effect on ADPR-capped RNAs.

3/ Line 183 – when discussing tRNAs capped with NAD, you likely mean pre-tRNAs as transcription of tRNA genes generates precursor tRNAs (pre-tRNAs) that need to be processed, e.g. trimmed at both termini, and modified to produce a mature, functional tRNA molecule. To avoid confusion, specify that the NAD-TSS are for pre-tRNAs.

Reviewer #2 (Remarks to the Author):

This revised manuscript, includes additional data data to support the authors conclusions that archaea RNA have NAD caps and that these RNAs are mainly dispensed in cells by a thermal conversion of the NAD to ADPR by the loss of nicotinamide. The resulting RNA is then substrate to a the *Saci-NudT5* nudix protein to yield a 5'monophosphate RNA that can then be degraded by the *Saci-aCPSF2* exonuclease. This is an interesting model, but the question whether this occurs in archaea cells or is a pathway observed in vitro still remains and was not addressed by the revision.

1. In response to original point #2, the authors tested the in vitro conversion of NAD-RNA to ADPR-RNA. However, this is not particularly informative and one would expect parameters in the cell to be considerably different than an in vitro reaction. For example, if a similar conversation of NAD to ADPR occurs in cells as it does in vitro, there probably would not be detectable NAD-RNA and not sure that it would have any function since it would be so labile. One would have to assume there are buffering systems in cells that maintain the NAD intact. At the natural high temperatures. Otherwise there would be no free NAD or NAD-RNA. Quantitative levels of of NAD-RNA and ADPR-RNA in cells is important.

In a related point (point #3 of original review), the authors respond that they addressed the cellular levels by testing the ratio of NAD to ADPR in cells with a 8:1 ratio in wild type cells and 2:1 ratio in the *Saci-aCPSF2* cells. They further state that 4.5 fold more NAD RNA results in the knockout strain. Collectively, this suggests that ADPR RNA levels remain the same in the two strains while NAD-RNA levels increase in the KO. These data do not support the premise of the manuscript that NAD-RNA is converted into ADPR-RNA and raise the likely possibility that the observations are restricted to in vitro. They do however implicate *Saci-aCPSF2* in the decapping of

NAD RNA which further detracts from the a main point of the manuscript that NAD RNA is primarily converted to ADPR to be degraded rather than directly degraded. The role of Scai-aCPSF2 should be further pursued (see below).

2. In response to my original point #4, the authors removed to offending figure and also provided in vitro data obtained from extract of wild type and KO cells. Although this is informative, it is difficult to make firm conclusions from extract. In light of my comments above, I still believe my original request for directly testing the activity of Scai-aCPSF2 is necessary. The fact that the protein forms in inclusion bodies and difficult to purify is not a satisfactory reason not to perform this critical experiment. There are multiple additional non bacterial expression systems with tagged protein expression in other organisms including mammalian cell if necessary or in vitro transitions systems that could be employed.

3. On line 244, the authors state the following: "the heat-treated NAD-RNA was used for an ADPR-decapping assay with hNudT5, which shows in vitro activity toward ADPR but not NAD-RNAs^{19,34}."

A quick perusal of these two references does not show assays of NudT5 against NAD-RNA. Munir et al (19) use Tpt1 with NAD to make 2'phosph-ADP-ribosylated RNA. They do not use NAD-RNA and I don't even see NudT5 used.

Zha et al (34) shows hNudT5 activity on ADPR. NAD-RNA is not used.

The authors should experimentally validate their statement and use a mutant hNudT5 to confirm any activity they detect is from the hNudT5 protein used.

4. In several instances the author mention that the NAD on NAD-RNA (31 min measured by the authors) is more stable than free NAD (24 min calculated in ref 14). These are two different experiments/parameters. Can they really be compared especially with such a minor difference? It seems to me a direct comparison in the same experiment would be the most accurate comparison and enable the authors to make a firm statement.

Reviewer #3 (Remarks to the Author):

Gomes-Filho et al. report the presence of NAD- and ADPR-capped transcripts in archaea and their susceptibility to attack by two archaeal enzymes, Scai-aCPSF2 and Scai_NudT5. The revised manuscript has been significantly improved by incorporating new data. Most importantly, the authors now present evidence, based on experiments with a Scai-aCPSF2 knockout strain, that this *S. acidocaldarius* exonuclease both degrades NAD-capped RNA in vitro and diminishes its concentration in vivo. It's a pity that they have not also determined directly whether the absence of Scai-aCPSF2 in that strain increases the lifetime of one or two representative NAD-capped RNAs in *S. acidocaldarius*, but the impact of the knockout mutation on the cellular concentration of NAD-capped RNA suggests that it does.

The authors have also obtained more evidence for the spontaneous conversion of NAD-capped RNA to ADPR-capped RNA at high temperatures. Unfortunately, they do not show all of this evidence. In particular, Figure 4d depicts only the time-dependent decrease in NAD-capped RNA at high temperature and not the corresponding increase in ADPR-capped RNA, which should be added to the graph. (Figure 4c merely shows ion chromatograms of chemically pure nicotinamide, NAD, and ADPR standards.)

Additional comments:

Lines 147-149. The evidence that NAD capping in these archaea occurs co-transcriptionally and not post-transcriptionally (coincidence of transcription start sites, untested promoter motifs) is very weak. To prove this point, the authors would need to show that changing the proposed promoter motifs alters the percentage of NAD-capped transcripts in these organisms. Without such evidence, this conclusion is not justified and should be described merely as a possibility.

Lines 169-184. These two paragraphs should be moved to the Discussion because they are largely speculative, especially the claim that capping most likely occurs co-transcriptionally.

Line 213. "Substituting ATP for NAD" should be changed to "Replacing ATP with NAD".

Lines 216-217. Please change "another pathway" to "an enzyme other than a Nudix hydrolase".

Line 280. This paragraph should be modified to explain how the evidence in Figure 5 comparing wild-type and Saci-aCPSF2 knockout strains "reinforces the ADPR-decapping activity in *S. acidocaldarius*".

Figure 5. In panel b, it is confusing to graph the fold change starting at a value of 0 (WT >> KO). It would be clearer either to use a log scale on the y-axis or to add a horizontal dashed line at a fold change of 1, which is actually no change at all. Furthermore, in panels c and e, it is not clear what was loaded in the lanes labeled NAD/5' p-RNA and ADPR/5' p-RNA. Finally, it is not clear what is meant in the figure legend by the term "absolute ratio". Is that the molar ratio?

Figure 6. The revised manuscript now describes two potential pathways by which NAD-capped transcripts can be degraded in *S. acidocaldarius*, one that involves exonucleolytic attack by Saci-aCPSF2 without prior decapping and another in which the spontaneous conversion of NAD caps to ADPR caps is followed by Scai_NudT5-mediated decapping and Saci-aCPSF2-mediated degradation. Oddly, although this central conclusion is illustrated in Fig. 6, it is never stated explicitly in the main text of the manuscript or even mentioned in the figure legend.

Response to reviewer comments

Reviewer #1 (Remarks to the Author):

The revised version is much improved. Overall, the two most exciting findings are: (i) *S. acidocaldarius* contains also ADPR (as a result of thermal degradation) at 5' ends of some RNAs and (ii) ADPR is removed by a dedicated nudix protein, Saci_NudT5. However, several key experiments are still missing, namely a direct proof of NAD-capping of selected RNAs and effects of Saci_NudT5 deletion/depletion on ADPR-capped RNAs. For details, see comments.

Comments:

1/ The authors have not confirmed the 5' end modifications of RNAs by NAD by a truly independent approach such as northern analysis using boronate gels. The RT-qPCR results are a good addition but still, they are based on an indirect approach. If you want to convince the field, perform northern blot analysis with boronate gels and RNA(s) that are abundant and NAD-capped. Alternatively (if it is the case), explain that the NAD-capped RNAs represent such a small fraction of the respective RNAs that it is not possible to detect them with boronate gels.

Response: We thank the reviewer for the suggestions. We performed the LC-MS/MS measurements, and the concentrations are now provided in the text (lines 210-216, lines 265-266) and the table below. We decided to include this data in the manuscript as the reviewer is correct that northern blot analysis of low abundance NAD-capped RNAs via boronate gels (APB gels) yielded rather faint and diffused signals for transcripts detected by NAD captureSeq, (*Sac_62_asRNA* and *Sac_6_k-turn*).

Table 1: Concentration of NAD and ADPR covalently linked to RNA in different *S. acidocaldarius* strains.

Strain	NAD (fmol / μg RNA)	ADPR (fmol / μg RNA)
S. acidocaldarius (MW001)	2 \pm 0.3	16 \pm 1
S. acidocaldarius Δ Saci-aCPSF2	8.7 \pm 0.6	20 \pm 1.2
S. acidocaldarius Δ Saci_NudT5	0.03 \pm 0.06	4.43 \pm 0.05

Line 277: You say "ADPR-RNA was processed into 5'-p-RNA at a slightly faster rate in the KO S30 extract (Fig. 5e-f). Altogether, these results suggest that the 5'-3' exonuclease Saci-aCPSF2 plays an essential role in the turnover of NAD-capped RNAs and reinforces the ADPR-decapping

activity in *S. acidocaldarius*.” If it reinforced ADPR decapping, the degradation rate of the Saci-aCPSF2 deletion strain should be slower than wt, not faster. The data actually suggest the opposite than you claim. Discuss/explain. Also, to support the main story, it would be highly desirable to create a knock-out of Saci_NudT5 (or at least a depletion strain) and test the effect on ADPR-capped RNAs.

Response: We thank the reviewer for the comments. In this paragraph, our objective was to point out that *S. acidocaldarius* S30 cell extracts present ADPR-decapping activity and not to imply that Saci-aCPSF2 is responsible for this process. We modified the text accordingly.

In a previous publication (Breuer et al. 2023) our group established a knockout strain for Saci_NudT5. Using this strain, we performed a new set of LC-MS/MS analyses and ADPR-decapping assays using S30 cell extracts. The Δ Saci_NudT5 strain showed 0.3 ± 0.06 fmol of NAD and 4.44 ± 0.5 of ADPR per μ g of RNA. This results in a fold-change of 0.2 ± 0.08 for NAD and 0.35 ± 0.03 for ADPR when comparing Δ Saci_NudT5 to the wild-type (Supp. Fig. 5a). Decapping assays using the S30 cell-extract revealed the, albeit slower, retention of ADPR-RNA decapping activities, strongly suggesting that other enzymes are also capable of performing this function. We expanded the text accordingly (lines 210-223) and the results were added to Supplementary Figure 5.

3/ Line 183 – when discussing tRNAs capped with NAD, you likely mean pre-tRNAs as transcription of tRNA genes generates precursor tRNAs (pre-tRNAs) that need to be processed, e.g. trimmed at both termini, and modified to produce a mature, functional tRNA molecule. To avoid confusion, specify that the NAD-TSS are for pre-tRNAs.

Response: The text was modified accordingly.

Reviewer #2 (Remarks to the Author):

This revised manuscript, includes additional data data to support the authors conclusions that archaea RNA have NAD caps and that these RNAs are mainly dispensed in cells by a thermal conversion of the NAD to ADPR by the loss of nicotinamide. The resulting RNA is then substrate to a the Scai-NudT5 nudix protein to yield a 5'monophosphate RNA that can then be degraded by the ScaiaCPDF2 exonuclease. This is an interesting model, but the question whether this occurs in archaea cells or is a pathway observed in vitro still remains and was not addressed by the revision.

1. In response to original point #2, the authors tested the in vitro conversion of NAD-RNA to ADPR-RNA. However, this is not particularly informative and one would expect parameters in the cell to be considerably different than an in vitro reaction. For example, if a similar conversation of NAD to ADPR occurs in cells as it does in vitro, there probably would not be detectable NAD-RNA and not sure that it would have any function since it would be so labile. One would have to assume there are buffering systems in cells that maintain the NAD intact. At the natural high temperatures. Otherwise there would be no free NAD or NAD-RNA. Quantitative levels of of NAD-RNA and ADPR-RNA in cells is important.

a) For example, if a similar conversation of NAD to ADPR occurs in cells as it does in vitro, there probably would not be detectable NAD-RNA and not sure that it would have any function since it would be so labile. One would have to assume there are buffering systems in cells that maintain the NAD intact. At the natural high temperatures. Otherwise there would be no free NAD or NAD-RNA.

Response: We thank the reviewer for the comments. The topic of handling NAD thermal degradation in thermophilic environments has been the focus of previous studies (Hachisuka et al. 2017, 2018). Overall, the most relevant finding is the presence of robust, and often redundant, pathways for the salvage of NAD following its thermal degradation. In this sense, ADPR-hydrolases are key enzymes in processing ADPR into substrates suitable for the NAD salvage pathway. Additionally, we discuss this issue in lines 230-235.

b) Quantitative levels of of NAD-RNA and ADPR-RNA in cells is important.

We thank the reviewer for the suggestions, we performed the LC-MS/MS measurements, and the concentrations are now provided in the text (lines 210-216, lines 265-266) and the table below.

Table 2: Concentration of NAD and ADPR covalently linked to RNA in different *S. acidocaldarius* strains.

Strain	NAD (fmol / μ g RNA)	ADPR (fmol / μ g RNA)
S. acidocaldarius (MW001)	2 \pm 0.3	16 \pm 1
S. acidocaldarius Δ Saci-aCPSF2	8.7 \pm 0.6	20 \pm 1.2
S. acidocaldarius Δ Saci_NudT5	0.3 \pm 0.06	4.43 \pm 0.05

In a related point (point #3 of original review), the authors respond that they addressed the cellular levels by testing the ratio of NAD to ADPR in cells with a 8:1 ratio in wild type cells and 2:1 ratio in the Saci-aCPSF2 cells. They further state that 4.5 fold more NAD RNA results in the knockout strain. Collectively, this suggests that ADPR RNA levels remain the same in the two strains while NAD-RNA levels increase in the KO. These data do not support the premise of the manuscript that NAD-RNA is converted into ADPR-RNA and raise the likely possibility that the observations are restricted to in vitro. They do however implicate Scai-aCPSF2 in the decapping of NAD RNA which further detracts from the a main point of the manuscript that NAD RNA is primarily converted to ADPR to be degraded rather than directly degraded. The role of Scai-aCPSF2 should be further pursued (see below).

Reviewer comment: *These data do not support the premise of the manuscript that NAD-RNA is converted into ADPR-RNA and raise the likely possibility that the observations are restricted to in vitro.*

Response: We agree that these data should not be used to reinforce the conversion of NAD-RNA to ADPR-RNA via thermal degradation in cells. As explored in lines 252-277, this analysis focuses on the effects of a Saci-aCPSF2 knockout on NAD- and ADPR-RNA levels. Moreover, the maintenance of ADPR-RNA levels could be explained by the fact that ADPR-decapping enzymes (e.g., Saci_NudT5) are still present and active in this genetic background.

Reviewer comment: *They do however implicate Scai-aCPSF2 in the decapping of NAD RNA which further detracts from the a main point of the manuscript that NAD RNA is primarily converted to ADPR to be degraded rather than directly degraded.*

Response: We agree with the reviewer that Scai-aCPSF2 is mainly implicated in the degradation of NAD-RNAs, as shown in Fig. 5a-d and discussed in lines 252-277. Concerning the point of NAD-RNA being primarily converted to ADPR-RNA via thermal degradation, we cannot objectively state that this is the main pathway, as other factors (e.g., NADases) are still unknown. Our objective is to propose an additional non-enzymatic pathway for the conversion of NAD to ADPR that could only be present in hyperthermophilic organisms. In our proposed model (Fig.6), Scai-aCPSF2 is already implicated as an NAD-RNA degrading enzyme.

2. In response to my original point #4, the authors removed the offending figure and also provided in vitro data obtained from extract of wild type and KO cells. Although this is informative, it is difficult to make firm conclusions from extract. In light of my comments above, I still believe my original request for directly testing the activity of Scai-aCPSF2 is necessary. The fact that the protein forms inclusion bodies and is difficult to purify is not a satisfactory reason not to perform this critical experiment. There are multiple additional non-bacterial expression systems with tagged protein expression in other organisms including mammalian cells if necessary or in vitro translation systems that could be employed.

Response: Following protein refolding, we obtained soluble recombinant wild-type enzymes and four mutant variants of Scai-aCPSF2. However, we still observed that maintaining protein solubility and stability remains challenging for different buffer conditions, temperature ranges and expression systems. Unfortunately, the research field lacks a complete understanding of 5'-to-3' exonuclease activities in Archaea. Available literature did not identify a catalytic mutant and indicates that other β -CASP proteins exist in *S. acidocaldarius* that were proposed to form a complex with Scai-aCPSF2 affecting mRNA turnover (Märtens, B. PLoS One. 2013; 8(10): e76569). We agree that it is desirable to sort this out but feel that this is beyond the scope of this manuscript. Therefore, we included cell extract assays coupled with LC-MS/MS quantification in our revised manuscript to ensure that all known and unknown archaeal mRNA degradation and decapping components are represented properly.

3. On line 244, the authors state the following: "the heat-treated NAD-RNA was used for an ADPR-decapping assay with hNudT5, which shows in vitro activity toward ADPR but not NAD-RNAs^{19,34}." A quick perusal of these two references does not show assays of NudT5 against NAD-RNA. Munir et al (19) use Tpt1 with NAD to make 2'-phospho-ADP-ribosylated RNA. They do not use NAD-RNA and I don't even see NudT5 used. Zha et al (34) shows hNudT5 activity on ADPR. NAD-RNA is not used. The authors should experimentally validate their statement and use a mutant hNudT5 to confirm any activity they detect is from the hNudT5 protein used.

Response: We thank the reviewer for pointing this out. The correct reference for that sentence was not number 19, but 18 instead (Abele F et al, A Novel NAD-RNA Decapping Pathway Discovered by Synthetic Light-Up NAD-RNAs. *Biomolecules*. 2020 Mar 28;10(4):513. doi: 10.3390/biom10040513). In this article, the authors test the activity of many putative decapping enzymes, including hNudT5, against NAD-RNAs. Figure 5b of this article provides evidence that “The human enzyme hNUDT5 hydrolyses only ^{Fur}NAD, but not ^{Fur}NAD-RNA despite its reported broad substrate specificity”.

4. In several instances the author mention that the NAD on NAD-RNA (31 min measured by the authors) is more stable than free NAD (24 min calculated in ref 14). These are two different experiments/parameters. Can they really be compared especially with such a minor difference? It seems to me a direct comparison in the same experiment would be the most accurate comparison and enable the authors to make a firm statement.

Response: We adjusted the text to indicate that these stability measurements are from different experiments.

Reviewer #3 (Remarks to the Author):

Gomes-Filho et al. report the presence of NAD- and ADPR-capped transcripts in archaea and their susceptibility to attack by two archaeal enzymes, Saci-aCPSF2 and Scai_NudT5. The revised manuscript has been significantly improved by incorporating new data. Most importantly, the authors now present evidence, based on experiments with a Saci-aCPSF2 knockout strain, that this *S. acidocaldarius* exonuclease both degrades NAD-capped RNA in vitro and diminishes its concentration in vivo. It's a pity that they have not also determined directly whether the absence of Saci-aCPSF2 in that strain increases the lifetime of one or two representative NAD-capped RNAs in *S. acidocaldarius*, but the impact of the knockout mutation on the cellular concentration of NAD-capped RNA suggests that it does.

The authors have also obtained more evidence for the spontaneous conversion of NAD-capped RNA to ADPR-capped RNA at high temperatures. Unfortunately, they do not show all of this evidence. In particular, Figure 4d depicts only the time-dependent decrease in NAD-capped RNA at high temperature and not the corresponding increase in ADPR-capped RNA, which should be added to the graph. (Figure 4c merely shows ion chromatograms of chemically pure nicotinamide, NAD, and ADPR standards.)

Response: We thank the reviewer for the comments. Figure 4c was changed accordingly and now demonstrates the ion chromatograms of NAD, ADPR, and Nm standards together with a heat-treated sample of NAD-RNA.

Additional comments:

Lines 147-149. The evidence that NAD capping in these archaea occurs co-transcriptionally and not post-transcriptionally (coincidence of transcription start sites, untested promoter motifs) is very weak. To prove this point, the authors would need to show that changing the proposed promoter motifs alters the percentage of NAD-capped transcripts in these organisms. Without such evidence, this conclusion is not justified and should be described merely as a possibility.

Response: We thank the reviewer for this suggestion. We adjusted our initial statement and modified the text accordingly (lines 146-148).

Lines 169-184. These two paragraphs should be moved to the Discussion because they are largely speculative, especially the claim that capping most likely occurs co-transcriptionally.

Response: The text was modified accordingly (lines 286-299)

Line 213. “Substituting ATP for NAD” should be changed to “Replacing ATP with NAD”.

Response: The text was modified accordingly.

Lines 216-217. Please change “another pathway” to “an enzyme other than a Nudix hydrolase”.

Response: The text was modified accordingly.

Line 280. This paragraph should be modified to explain how the evidence in Figure 5 comparing wild-type and Saci-aCPSF2 knockout strains “reinforces the ADPR-decapping activity in *S. acidocaldarius*”.

Response: We thank the reviewer for pointing out this. In this paragraph, our objective was to highlight that *S. acidocaldarius* S30 cell extracts present ADPR-decapping activity and not to imply that Saci-aCPSF2 is responsible for this process. We modified the text accordingly.

Figure 5. In panel b, it is confusing to graph the fold change starting at a value of 0 (WT >> KO). It would be clearer either to use a log scale on the y-axis or to add a horizontal dashed line at a fold change of 1, which is actually no change at all. Furthermore, in panels c and e, it is not clear what was loaded in the lanes labeled NAD/5' p-RNA and ADPR/5' p-RNA. Finally, it is not clear what is meant in the figure legend by the term “absolute ratio”. Is that the molar ratio?

Response: We thank the reviewer for the suggestions. The figure was changed accordingly, and the legend was adjusted.

Figure 6. The revised manuscript now describes two potential pathways by which NAD-capped transcripts can be degraded in *S. acidocaldarius*, one that involves exonucleolytic attack by Saci-aCPSF2 without prior decapping and another in which the spontaneous conversion of NAD caps to ADPR caps is followed by Scai_NudT5-mediated decapping and Saci-aCPSF2-mediated degradation. Oddly, although this central conclusion is illustrated in Fig. 6, it is never stated explicitly in the main text of the manuscript or even mentioned in the figure legend.

Response: We thank the reviewer for this suggestion. We modified the text (lines 321-326) and the legend of Figure 6 accordingly.

Reviewer #1 (Remarks to the Author):

The manuscript "Identification of NAD-RNAs species and ADPR-RNA decapping in Archaea" is almost OK from my point of view. Nevertheless, I strongly suggest to be totally transparent and specify in the text (e. g. in Discussion) that APB gels coupled with northern analysis were used to detect the modified RNAs but failed due to the low level of this modification in individual RNAs.

Reviewer #2 (Remarks to the Author):

The revised report with the toned down in vivo assertions have improved the report significantly. Just two minor suggestions.

1. What does "reinforces the need of salvage pathways as described for free NAD" on line 250 mean? Please elaborate.
2. I do not see Table 1 in the manuscript, only in the rebuttal letter. This should be included in the manuscript.

Reviewer #3 (Remarks to the Author):

This new revision of the manuscript by Gomes-Filho et al. is better than before, but one of my principal criticisms has not yet been addressed. To demonstrate that NAD-capped RNA is spontaneously converted to ADPR-capped RNA at high temperatures, the authors need to show not only a time-dependent decrease in NAD-capped RNA but also a corresponding increase in ADPR-capped RNA. Figure 4 still shows only the decrease in NAD-capped RNA.

In addition, I'm confused by one of the new results that has been added to the revised manuscript (lines 213-223 and Supplementary Figure 5). Namely, in view of the authors' finding that Saci_NudT5 has ADPR- but not NAD-decapping activity, it seems counterintuitive that knocking out the Saci_NudT5 gene significantly reduces the amount of both ADPR- and NAD-capped RNA in *S. acidocaldarius*. The authors should explain how the absence of this ADPR-decapping enzyme decreases (rather than increases) the cellular concentration of ADPR-capped RNA and why its absence has any effect at all on the concentration of NAD-capped RNA. The possible presence of additional enzymes involved in ADPR decapping (line 223) could have been invoked as an explanation if knocking out Saci_NudT5 caused little or no increase in the concentration of ADPR-capped RNA, but it is not clear how this hypothesis can explain the substantial reduction in ADPR-capped RNA that is observed.

RESPONSE TO THE REVIEWER COMMENTS

Reviewer #1 (Remarks to the Author):

The manuscript "Identification of NAD-RNAs species and ADPR-RNA decapping in Archaea" is almost OK from my point of view. Nevertheless, I strongly suggest to be totally transparent and specify in the text (e. g. in Discussion) that APB gels coupled with northern analysis were used to detect the modified RNAs but failed due to the low level of this modification in individual RNAs.

We thank the reviewer for the suggestion. The results of the northern blot are now discussed in lines 120-124.

Reviewer #2 (Remarks to the Author):

The revised report with the toned down in vivo assertions have improved the report significantly. Just two minor suggestions.

1. What does "reinforces the need of salvage pathways as described for free NAD" on line 250 mean? Please elaborate.

Our results suggest that NAD covalently linked to RNA remains heat-labile generating ADPR and Nicotinamide after heat degradation. Further degradation steps (e.g., exonucleolytic activity) would release free ADPR in the cell that requires enzymes such as Saci_NudT5 to be converted. This paragraph was rewritten to avoid confusion (lines 252-254).

2. I do not see Table 1 in the manuscript, only in the rebuttal letter. This should be included in the manuscript.

The results are now provided in Supplementary Table 6.

Reviewer #3 (Remarks to the Author):

This new revision of the manuscript by Gomes-Filho et al. is better than before, but one of my principal criticisms has not yet been addressed. To demonstrate that NAD-capped RNA is spontaneously converted to ADPR-capped RNA at high temperatures, the authors need to show not only a time-dependent decrease in NAD-capped RNA but also a corresponding increase in ADPR-capped RNA. Figure 4 still shows only the decrease in NAD-capped RNA.

We modified Figure 4d to show the concurrent increase of ADPR. Additionally, the experiments shown in Figure 4a-b (heat treatment followed by hNudT5 ADPR-decapping and APB-polyacrylamide gels) evidenced the conversion of NAD-RNAs to ADPR-RNAs in a time-dependent manner.

In addition, I'm confused by one of the new results that has been added to the revised manuscript (lines 213-223 and Supplementary Figure 5). Namely, in view of the authors' finding that Saci_NudT5 has ADPR- but not NAD-decapping activity, it seems counterintuitive that knocking out the Saci_NudT5 gene significantly reduces the amount of both ADPR- and NAD-

capped RNA in *S. acidocaldarius*. The authors should explain how the absence of this ADPR-decapping enzyme decreases (rather than increases) the cellular concentration of ADPR-capped RNA and why its absence has any effect at all on the concentration of NAD-capped RNA. The possible presence of additional enzymes involved in ADPR decapping (line 223) could have been invoked as an explanation if knocking out *Saci_NudT5* caused little or no increase in the concentration of ADPR-capped RNA, but it is not clear how this hypothesis can explain the substantial reduction in ADPR-capped RNA that is observed.

ADPR-hydrolases (such as *Saci_NudT5*) are proposed to be key enzymes in the NAD salvage pathway, which is critical for hyperthermophilic organisms to maintain proper amounts of this metabolite. The knockout of *Saci_NudT5* disrupts this pathway and impacts intracellular levels of free NAD. Lower concentration of free NAD would also suppress co-transcriptional NAD-capping. ADPR-RNAs are proposed to be generated post-transcriptionally either by an unknown NADase or by heat-degradation acting on NAD-RNAs. Consequently, ADPR-RNA levels would be directly dependent on the incorporation of NAD into nascent RNAs. We added a brief discussion on this subject to the text (lines 326-332).

Moreover, although the absolute levels of NAD- and ADPR-RNAs are lower, the ADPR-RNA/NAD-RNA ratio is significantly higher in the Δ *Saci_NudT5* strain (15:1) when compared to the wild-type (8:1), which we believe points to less efficient decapping of the remaining ADPR-RNAs in the Δ *Saci_NudT5* strain.